# Multifunctional flexible membranes from sponge-like porous carbon nanofibers with high conductivity

Jianhua Yan [1,2], Keqi Dong[1], Yuanyuan Zhang[1], Xiao Wang[3], Ahmed Abdulqawy Aboalhassan[3], Jianyong Yu[2] & Bin Ding[1,2]*

Conductive porous carbon nanofibers are promising for environmental, energy, and catalysis applications. However, increasing their porosity and conductivity simultaneously remains challenging. Here we report chemical crosslinking electrospinning, a macro–micro dual-phase separation method, to synthesize continuous porous carbon nanofibers with ultrahigh porosity of >80% and outstanding conductivity of 980 S cm$^{-1}$. With boric acid as the crosslinking agent, poly(tetrafluoroethylene) and poly(vinyl alcohol) are crosslinked together to form water-sol webs, which are then electrospun into fibrous films. After oxidation and pyrolysis, the as-spun fibers are converted into B-F-N triply doped porous carbon nanofibers with well-controlled macro–meso–micro pores and large surface areas of ~750 m$^2$ g$^{-1}$. The sponge-like porous carbon nanofibers with substantially reduced mass transfer resistances exhibit multifunction in terms of gas adsorption, sewage disposal, liquid storage, supercapacitors, and batteries. The reported approach allows green synthesis of high-performance porous carbon nanofibers as a new platform material for numerous applications.

[1] State Key Laboratory for Modification of Chemical Fibers and Polymer Materials, College of Textile, Donghua University, Shanghai 201620, China. [2] Innovation Center for Textile Science and Technology, Donghua University, Shanghai 200051, China. [3] College of Materials Science and Engineering, Donghua University, Shanghai 201620, China. *email: binding@dhu.edu.cn

Conductive carbon with hierarchical porous structures enable fast access by electrons, ions, molecules, and particles[1]. Energy-storage devices, dye and gas adsorption apparatus, liquid storage facilities, and filters are examples of the rapidly growing applications of porous carbon materials[2]. In comparison with particles, fibers have greater development prospects, as they are easier to construct freestanding 2D films without using binders, which introduce additional interfacial resistances and are detrimental to rapid mass transfer[3]. Especially when the diameter of fibers decreases from micro to nanoscale, the diameter refinement endowed carbon nanofibers (CNFs) unique thermal and electrical properties, making them popular in various fields such as environmental and energy applications[4,5].

Extensive research has been focused on fabricating porous CNFs (PCNFs), but increasing porosity and conductivity of CNFs simultaneously is regarded as an open problem, as conductivity is generally inversely proportional to pore volumes[6–13]. On one hand, traditional methods for making PCNFs are somewhat intricate and unsafe[8–12]. Moreover, to ensure the structural integrity, most reported PCNFs only contain meso and micro pores with a low porosity of <20%[6–13]. PCNFs are generally fabricated by treating CNFs via activation or by designing CNF precursors that contain polymer templates and pore inducers[6,7]. The main route of activation is to create pores by etching CNFs with corrosive chemical agents such as KOH or $HNO_3$[8,9]. However, there are technical, environmental, and safety issues in their industrial production[10]. For the template method, there are no safety hazard problems, but conventional syntheses rely on blending polyacrylonitrile (PAN) or pitch with sacrificial polymers, which results in poorly controlled pores[11,12].

On the other hand, the conductivity of PCNFs is essential when applying them as freestanding electrodes or supporting scaffolds[14]. The concept of freestanding electrode has been reported in a number of papers[15–18]. However, this concept is not practical to advance into commercial manufacturing, as most of these PCNFs have low conductivity of <20 S cm$^{-1}$ [15–18]. Moreover, sheet resistances of such electrodes will be higher once the electrode areas are increased (i.e., tens of cm$^2$)[18]. This issue is concealed in lab-scale electrodes with limited areas. In particular, the conductivity is generally inversely proportional to pore volumes[15]. Therefore, increasing the porosity and conductivity of PCNFs simultaneously while maintaining their integrity is still a great challenge.

Here we develop chemical crosslinking electrospinning, a macro–micro dual-phase separation method, to create flexible and highly conductive PCNFs with large pore volumes. Having noticed that the mixture uniformity of the carbon precursor and the pore inducer was essential to control the porosity in CNFs, we selected poly (tetrafluoroethylene) nanoparticles (PTFE NPs), poly (vinyl alcohol) (PVA), and boric acid (BA) as the pore inducer, carbon precursor, and crosslinking agent, respectively. BA can react with PVA by chemical complexation and bond PTFE via hydrogen-bonding crosslinking[19,20]. Therefore, the stable sol of PVA–BA–PTFE can be prepared and electrospinning is carried out without producing entangled PVA macromolecules, which promises to enhance the continuity of the as-spun fibers. During the subsequent oxidation at 280 °C, the PVA dehydrogenated and formed conjugated C=C bonds, enhancing the stability of the fibers[21]. After pyrolysis in $N_2$ atmosphere at high temperatures of 800–1200 °C, the oxidized fibers were converted to B, F, and N triply doped PCNFs with interconnected macro (~68 nm), meso (~35 nm), and micro pores (~0.5 nm).

Benefiting from the hierarchical porous structures, the sponge-like PCNFs have high porosity of >80% and large surface areas of ~750 m$^2$ g$^{-1}$. On the other hand, the synergistic B–F–N doping effects endowed the freestanding PCNF films with a high conductivity of over 980 S cm$^{-1}$. Such PCNFs facilitated rapid matter transfer and exhibited multifunction. For example, they showed a leading level of $CO_2$ adsorption capacity of 3.9 mmol g$^{-1}$, a high mass ratio of 62 for storing ethylene glycol (EG), and could rapidly absorb methylene blue (MB) with a large capacity of 2250 mg g$^{-1}$, showing high performance in environmental management. Moreover, the all-carbon supercapacitors delivered a high power density of 1.75 kw kg$^{-1}$ and a large energy density of 42.77 Wh kg$^{-1}$ at 1 A g$^{-1}$, and the Li-sulfur batteries exhibited a high discharge capacity of >1000 mA h g$^{-1}$ at 1 C over 300 cycles, displaying great potential in developing high-performance energy-storage devices.

## Results

**Synthesis of B–N–F triply doped sponge-like PCNFs**. Figure 1a proposes a general overview of the synthesis of PCNFs using electrospinning followed by oxidation and pyrolysis. Starting from preparing homogeneous spinning sol of PVA-BA-PTFE, we first investigated the dynamic complexing reactions during the solation process. PVA was easily self-crosslinked due to the hydrogen bonding between/in the linear PVA molecules[19]. However, when introducing BA into this system, a uniform crosslinking sol network could be easily formed, as the BA not only chemically reacted with PVA to generate borate diol bonds, but also provided empty electron orbitals to coordinately bond PTFE (Supplementary Fig. 1)[22,23]. The hydrogen bonding between PTFE and PVA at the molecular level was regarded as an end-capping agent for regulating the crosslinking degree of PVA and PTFE. The sol was then electrospun into white fibrous films with an area of 70 × 55 cm$^2$ (Fig. 1b). Both the borate diol bonds and hydrogen bonds were dynamically reversible when they were stretched during the electrospinning, facilitating the formation of stable fibers of PVA@PTFE with uniform morphology (Fig. 1c)[24].

To prevent PVA from decomposing directly during high-temperature pyrolysis, it was necessary to stabilize the as-spun fibers by oxidation. In general, PVA melts at ~150 °C, decomposes at ~200 °C, and dehydrogenates at ~250 °C[25]. However, if we properly control the mass ratio (here, 5:5 was used) of PVA and PTFE, the as-spun fibers could withstand 280 °C due to the chemical and hydrogen bonds. At this temperature, the PVA had a small viscosity of 2794.5 cp and could self-assemble into a film (Supplementary Fig. 2). This property enabled the PVA with a good contact with PTFE, as verified by the bulging PTFE NPs and the rough surfaces of the as-spun fibers (Fig. 1d). After oxidation, the white film turned to brown (Supplementary Fig. 3) but the fiber diameters did not change much (Supplementary Fig. 4), indicating little decomposition of PVA. A possible change was the formation of conjugated C=C bonds, as shown in the Fourier transform infrared spectroscopy (FTIR; Supplementary Fig. 5). If the pre-oxidation process was skipped, the as-spun fibers could not transform into PCNFs.

During the subsequent pyrolysis in $N_2$ at 1200 °C, the thermally annealed fibers were converted into CNFs and, simultaneously, the micro and macrophase separations brought a homogeneous pore distribution. Small-molecules (BA and PVA) pyrolysis created meso and micro pores, whereas the decomposition of large PTFE NPs left continuous macropores (Fig. 1e). After pyrolysis, the small diameter change of the fibers indicated highly porous characteristics of the PCNFs. The chemical crosslinking between PTFE and PVA enhanced the carbon yield, as checked by the thermogravimetric (Supplementary Fig. 6) analysis. Both the PVA and PTFE could transform into CNFs and the total carbon yield was ~4.75%. The PCNFs displayed a mixed graphitic and amorphous structure (Supplementary Fig. 7) with a large porosity of >80%. According to

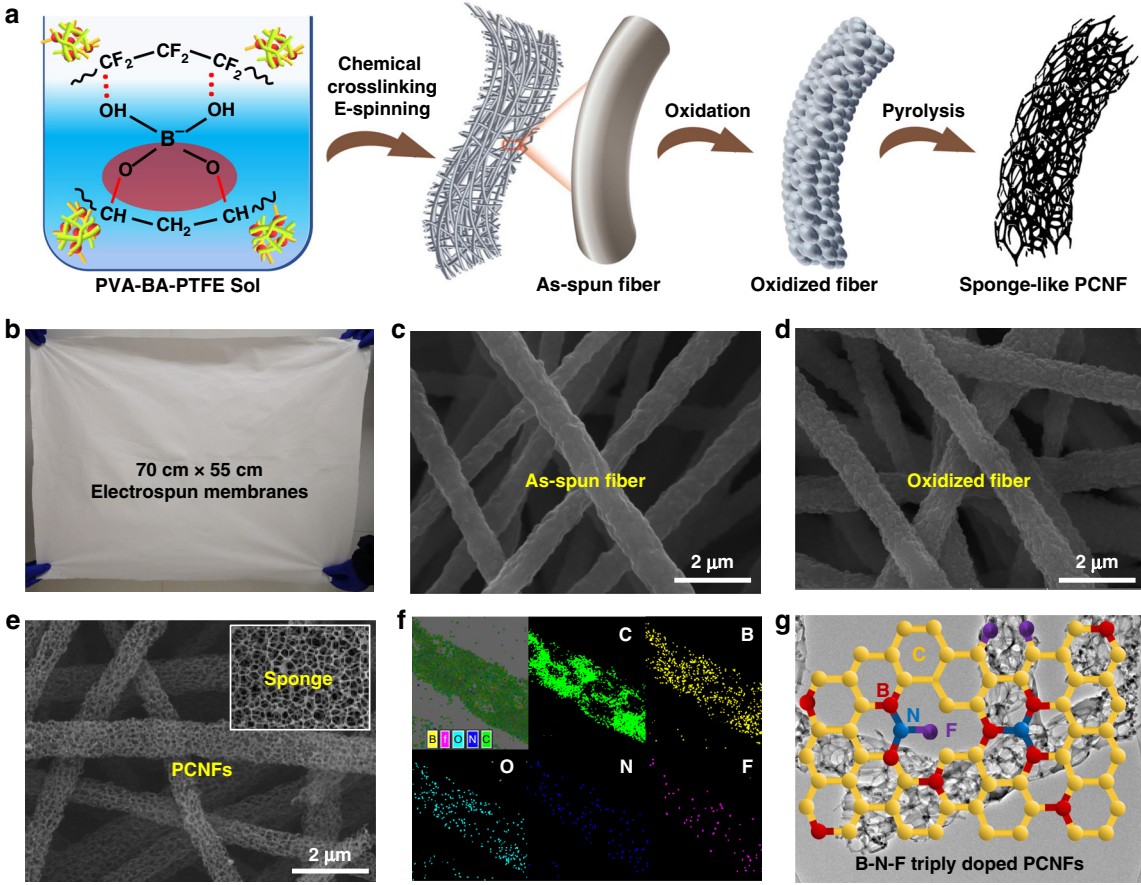

**Fig. 1 Fabrication of B–N–F triply doped sponge-like PCNFs. a** A general picture of using the chemical crosslinking electrospinning method to synthesize PCNFs. **b** A digital photo of the as-spun film with a size of 70 cm × 55 cm. **c–e** Scanning electronic microscopy (SEM) images of the as-spun fibers, the oxidized fibers and the PCNFs. **f** EDS mapping spectrum of PCNFs. **g** The proposed chemical model of B-N-F doped PCNFs.

energy dispersive spectrometer mapping, the PCNFs contained B, F, and N (Fig. 1f), where the N was probably caused by the reaction of $N_2$ with B during the high-temperature pyrolysis. The existence of B, F, and N in the PCNFs was further examined by X-ray photoelectron spectroscopy (Supplementary Fig. 8), in which the atomic percents of C, O, N, B, and F were 94.33%, 3.15%, 1.25%, 0.93%, and 0.34%, respectively. According to these results, the proposed chemical model of such triply doped PCNFs is shown in Fig. 1g.

Of note, the morphology of the PCNFs were greatly influenced by BA additive, the mass ratios of PVA and PTFE, and the pyrolysis temperatures. Without using BA, the PCNFs were not straight (Supplementary Fig. 9), which were prone to fracture. On the other hand, to obtain continuous PCNFs, the largest ratio of PVA and PTFE was 7:3. PTFE could prevent the diffusion of PVA at high temperatures, enhancing the stability of the PCNFs. If increasing the ratio to 8:2 or above, PCNFs would no longer be formed after pyrolysis (Supplementary Fig. 10a), whereas decreasing the ratios to 5:5 and 3:7 led to increased porosity (Supplementary Fig. 10b–d) and decreased diameters (Supplementary Fig. 11). Of note, from the scanning electronic microscopy (SEM) images, the macropore sizes of these different PCNFs look similar, indicating that the continuous macropores were from the decomposition of PTFE. In addition, a high pyrolysis temperature always led to small diameter and large porosity of PCNFs (Supplementary Figs. 12 and 13).

**Porosity characterization and conductivity analysis of PCNF membranes.** The thickness of a typical freestanding PCNF film

was 210 μm (Fig. 2a). The film had an intact structure, in which the PCNFs entangled with each other that offered a high degree of interconnectivity. These PCNFs had an average diameter of 700 nm, as shown by SEM (Fig. 2b and Supplementary Fig. 14). The pore distribution was revealed by the contrast between the carbon domains and the porous structures from the transmission electron microscopy (TEM; Fig. 2c), in which the macropores were distributed throughout the NFs. The pore size was measured to be ~60 nm by the high-magnification TEM (Fig. 2d). There were a mixture of graphitic, turbostratic, and amorphous carbon domains in the PCNFs, in which layered crystal structures with an average inter-planar distance of ~0.37 nm were observed (Supplementary Fig. 15)[26]. The freestanding PCNF films demonstrated excellent shape-memory performance and they could maintain their original shapes without cracking after releasing the applied bending forces (Fig. 2e).

The specific surface areas and the pore types of the PCNFs were checked by $N_2$ adsorption–desorption measurements (Fig. 2f). The PCNFs presented type I and type IV isotherm characteristics, showing hierarchical porous structures. The hysteresis loops of the adsorption–desorption branches did not coincide absolutely, indicating a heterogeneous pore size distribution in the PCNFs[27]. When $P/P_0$ was <0.1 and >0.9, the $N_2$ absorption increased sharply, which indicated that there were large amounts of micro–meso and macropores in the PCNFs, respectively[28]. The pore size-distribution curves further confirmed the existence of a large number of macro, meso, and micro pores in the PCNFs (Fig. 2g). The micro pore size was mainly concentrated at 0.5 nm, whereas the mesopore size was at 35 nm

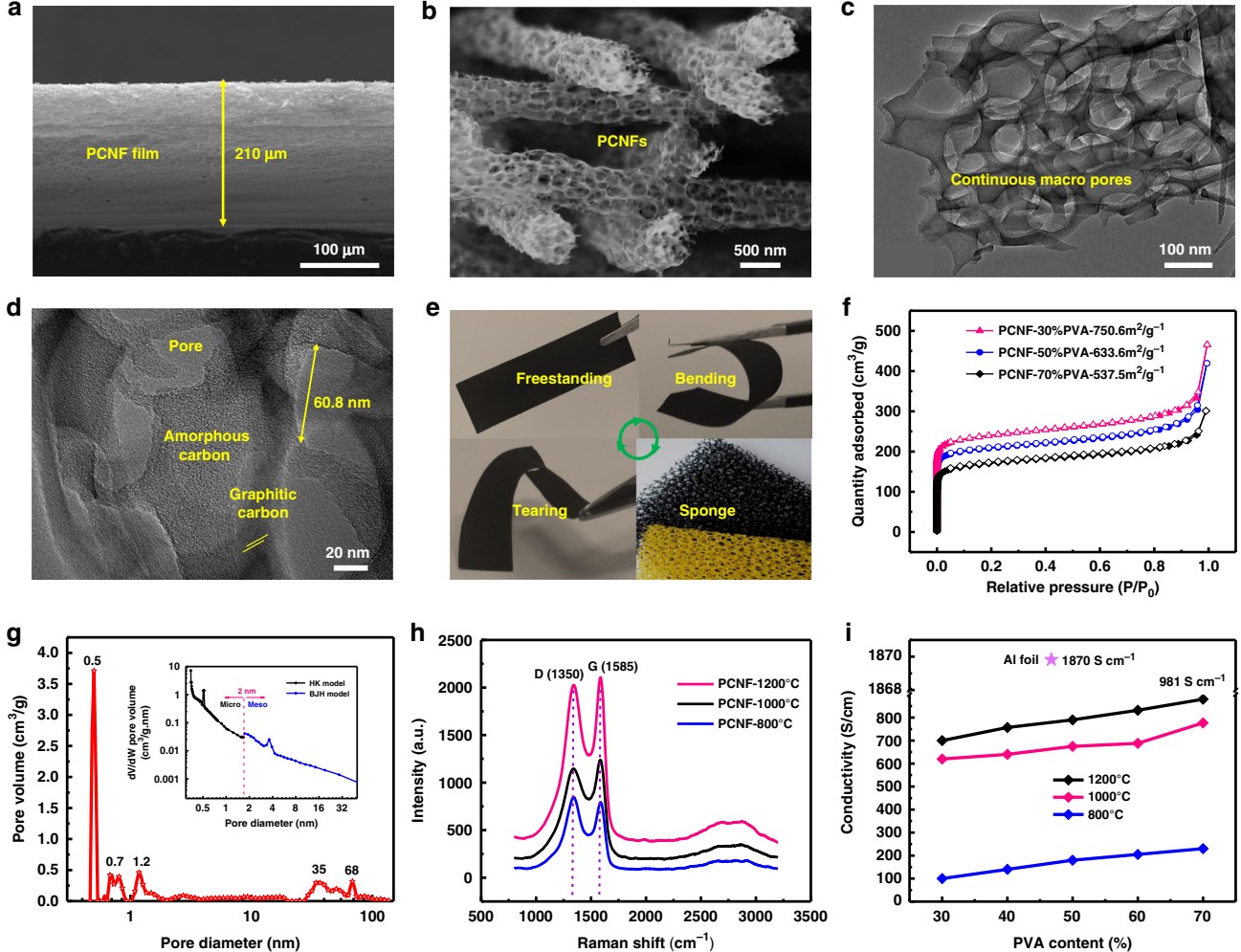

**Fig. 2 Characterizations of PCNF films. a** A cross-sectional SEM image of the PCNF film with a thickness of 210 μm. **b** A high-magnification SEM image of the PCNFs with continuous macropores. **c, d** TEM images of a single PCNF. **e** Demonstration of the robustness of the freestanding PCNF films. **f, g** N$_2$ adsorption–desorption isotherms of PCNFs. **h** Raman spectra of PCNFs with different pyrolysis temperatures. **i** Conductivity of PCNFs that produced different pyrolysis temperatures and PVA contents. Source data are provided as a Source Data file.

and macro pore size was at 68 nm, which was consistent with the roughly measured value by TEM. The insert figure expanded the distribution of micro and meso pores in different models. Supplementary Table 1 compared the surface areas of PCNFs produced with different polymer contents at 1200 °C. The perfect match of the pore size distributions of the different PCNFs (produced with different mass ratios of PVA and PTFE) in Supplementary Fig. 16 verified the feasibility of our method for controlling the pore sizes. Compared with PCNF–30% PVA and PCNF–70% PVA, the PCNF–50% PVA had the largest surface areas of 750.6 m$^2$ g$^{-1}$ (as calculated in the linear range of $P/P_0 =$ 0.01 to 0.1) with the largest total pore volume of 0.58 cm$^3$ g$^{-1}$ and mesopore volume of 0.42 cm$^3$ g$^{-1}$.

The graphitization degree and the defect characteristic of the PCNFs were checked by Raman spectra, as shown in Fig. 2h. The D-band (1355 cm$^{-1}$) and G-band (1582 cm$^{-1}$) represented the defected or disordered carbon domains and the ordered graphite crystalline structures, respectively, although the peak appearing between 2500 and 3000 cm$^{-1}$ was from the second order of the D-band, indicating the layered carbon structures[29]. When increasing the pyrolysis temperatures from 800 to 1000 °C, the defect degree (calculated by the ratio of peaks' intensities, $R_D = I_D/I_G$) decreased due to the increased graphite microcrystals in the planes, which enhanced the conductivity of the PCNFs

(Fig. 2i). When further increasing the temperature to 1200 °C, the graphite microcrystals changed from the disordered two-dimensional to ordered three-dimensional (3-D) structures, which enlarged the non-localized region of the large π-bonds and thus further enhanced the conductivity[30]. The conductivity of the PCNFs could reach as high as 981 S cm$^{-1}$, which was two to three orders of magnitude higher than the normal freestanding CNF films (<20 S cm$^{-1}$). The high conductivity was comparable with the commercial aluminum (Al) foils for battery current collectors, which had an electronic conductivity of 1870 S cm$^{-1}$, as tested by the same method (Supplementary Fig. 17).

**Practical applications of the PCNF films.** Flexible PCNFs with high surface areas and hierarchical macro–meso–microporous structures enabled outstanding matter storage capability and provided efficient pathways for rapid matter transportation (Supplementary Fig. 18). However, for practical applications, the mechanical properties of such PCNF films are essential. Figure 3a presents a typical tensile stress–strain curve of the PCNF film that fabricated at 1200 °C, which delivered a high Young's modulus of 429 MPa with a mechanical strength of 2.23 MPa and an extended strain of 0.52% before failure. In addition, the PCNFs exhibited a bending rigidity of only 13.1 mN, which was much

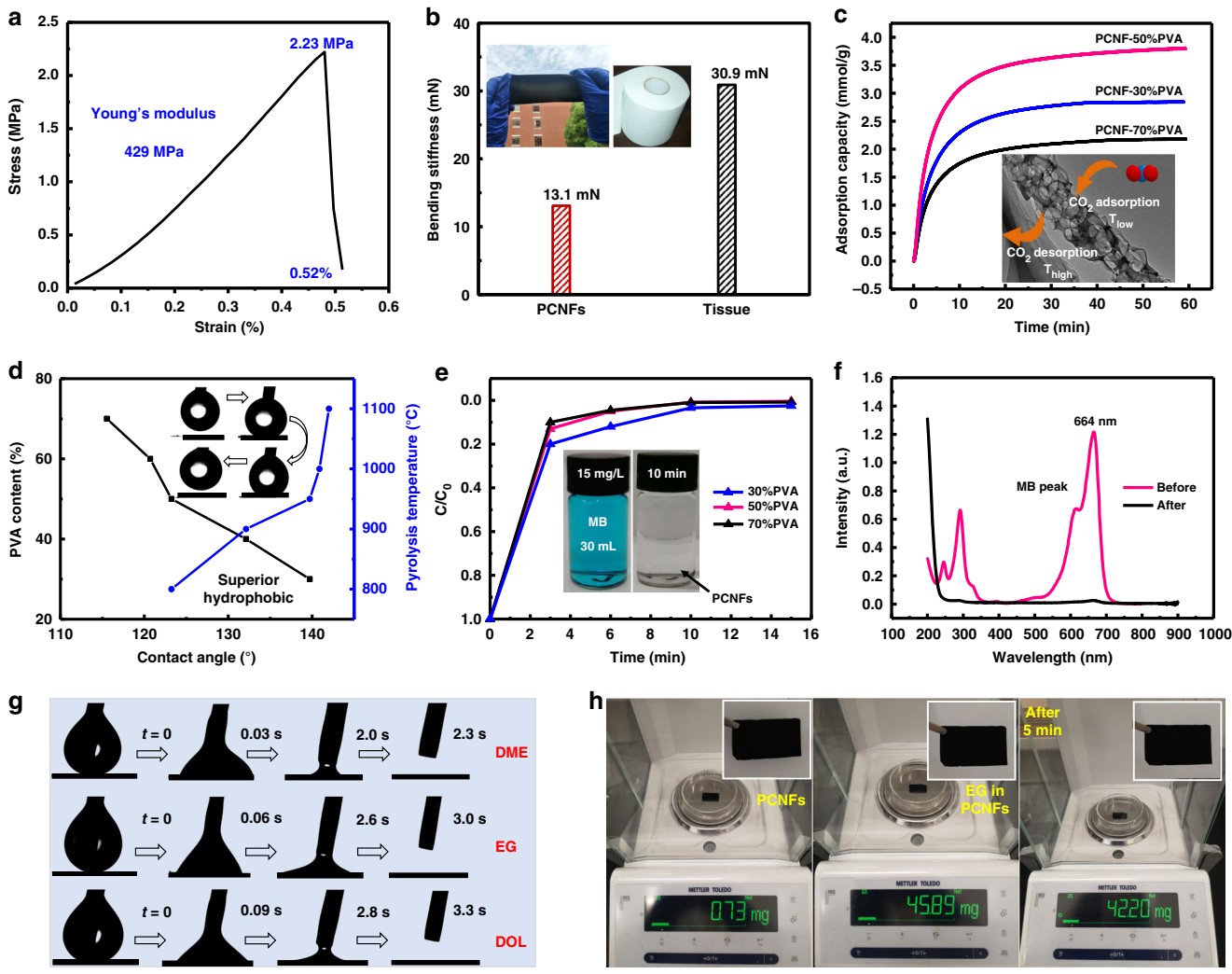

**Fig. 3 Practical applications of the PCNF films. a** Tensile stress–strain curve of the PCNF–50% PVA film that fabricated at 1200 °C. **b** Comparison of the bending rigidity between the PCNF film and the tissue. **c–e** The $CO_2$ adsorption performance, super hydrophobic properties, and dye-absorption properties of the PCNFs. **f** Characterization of ultraviolet spectrum of using the PCNFs to absorb the MB dye. **g** The organic solvent wettability and **h** liquid storage performance of the PCNFs. Source data are provided as a Source Data file.

smaller than the value of a paper towel (Fig. 3b). The smaller the bending stiffness, the better the softness. Such mechanical performance made it possible to apply these PCNFs into real applications for polluted gas adsorption, sewage disposal, liquid storage, energy storage, and so on.

Over-emission of greenhouse gases has become one of the main killers that endanger public health and the world economy[31]. In this sense, $CO_2$ capture have received extensive attentions. Conventional solid adsorbents such as silica gel and calcium lime that were used for $CO_2$ adsorption usually had inefficient and poor stability problems[4,32]. On the contrary, the PCNFs had abundant pore structures and good chemical stability. In addition, the capillary effect in the nanofibrous membranes render them a smaller adsorption resistance than the granular solid adsorbents. The $CO_2$ uptake behavior of the PCNFs was evaluated by thermogravimetry analysis (TGA, Fig. 3c). Three different PCNF membranes were kept in flowing $CO_2$ at 25 °C for adsorption and then were desorbed in flowing $N_2$ at 105 °C. As the PCNF–50% PVA had the largest surface areas, it delivered the highest $CO_2$ uptake capacity of 3.9 mmol g$^{-1}$, which is currently at the leading level in comparison with the reported ones[33]. In addition, the $CO_2$ diffused quickly in the PCNFs, as verified by the short time (<20 min) to reach the nearly saturation point. The

macropores with large inner surface areas were favorable for rapid caging $CO_2$ molecules, which were then adsorbed by the meso- and micro pores by strong surface adsorption. The high-efficient adsorption and diffusion features provided good prospects for applying the PCNFs to capture $CO_2$ and such films could be used in conjunction with the wearable products because of their light weights and flexibility.

In addition to gas adsorption, the PCNFs could be used as a promising adsorbent for organic pollutant absorption. The printing and dyeing industry has serious dye pollution, but it's difficult to treat the macromolecular organic dyes with deep pigmentation. Activated carbon and oxidizing agents have been developed to tackle this issue[34,35]. However, limited by small pore sizes, the activated carbon could only adsorb small molecular dyes. On the other hand, the oxidizing agents such as ozone could decompose the dyes into small substances to lose their chromogenic ability, but it's easy to cause secondary pollution. Here we use MB water solution to evaluate the adsorption capability of the PCNFs.

The PCNF films (pyrolysis at 1100 °C) were hydrophobic with a contact angle as high as 140.5°, as shown in Fig. 3d. According to the Cassie model, PCNF films with such a high porosity and nanoscale rough surfaces could not permeate the water droplets,

which were objects with clearly defined surface and surface tension[36]. When they contact, the air would absorb the voids of the nano-grooves and formed "air cushion" between the droplets and the nanostructure surface, effectively preventing the water droplets from infiltrating. On the other hand, F was regarded as an hydrophobic agent and the F-containing polymers were always applied for fabricating waterproof textiles[37,38]. Of note, for testing the contact angles, the highest pyrolysis temperature was 1100 °C. At 1200 °C, the PCNF films presented superior hydrophobic property, as shown in the inset of Fig. 3c, which depicted the process of a water droplet from touching to lifting off the PCNF surfaces. When immersed a small piece of PCNF film, with a weight of 2 mg, into the MB solution (15 mg L$^{-1}$, 30 mL) and stirred at room temperature, it could adsorb MB in 10 min (Fig. 3e, f), showing a large absorption rate. Of note, even when the weight of the film was reduced to 0.2 mg, it could also adsorb MB completely in 45 min, exhibiting a high adsorption capacity of 2250 mg g$^{-1}$. The hierarchical pores enlarged the internal surface areas for deep MB adsorption and enhanced the adsorption stability. In addition, the interactions between the PCNFs and the conjugated aromatic skeleton of MB formed stable $\pi$–$\pi$ stacking, which were beneficial for MB adsorption[39]. It should be noted that as macropores were not optimal for capturing dye molecules, the PCNF–30% PVA with much macropores had the lowest absorption efficiency.

Although the PCNFs were hydrophobic, they demonstrated excellent organic liquid wettability due to the rich surface functional groups. A series of dynamic processes including contact, attachment, infiltration, and separation of droplets on the PCNF films were recorded by a high-speed camera to observe the wettability (Fig. 3g). When adding the liquids onto the PCNF films, the droplets spread out rapidly and were completely absorbed. The hierarchical porous structure reduced the resistance and direction selectivity of the molecule movements. Moreover, it is believed that the heteroatom-doped PCNFs had strong interactions with the polar surface groups of the organic solvents, enabling a fast solvent wicking speed[40]. With such property, the PCNFs with large pore volumes could be used as liquid storage materials. As shown in Fig. 3h, the PCNFs delivered a high mass ratio of 62 for storing EG without leaking. The storing performance for other organic liquids were shown in Supplementary Fig. 19.

The development of efficient energy-storage techniques is believed as an effective way to solve the worldwide problem of energy crisis[41]. With the exception of the conspicuous sponge-like porous structures with high surface areas that were beneficial for mass transfer, the freestanding PCNF films offered high conductivity. These features rendered the PCNF electrodes capable of rapid storing a large amount of electrochemical energy. The direct use of self-supporting PCNFs as electrodes or scaffolds remove the need for loading binders or conductive additives, which produce large interfacial resistances and are detrimental to fast charge and discharge[42]. To evaluate the electrochemical performance of the PCNFs, we first assembled all-carbon supercapacitors containing symmetric PCNF electrodes and used the commercial Li-ion electrolytes as the supercapacitor electrolytes.

The ion and charge-transport dynamics were revealed by electrochemical impedance spectroscopy (EIS, Fig. 4a). In the Nyquist plots, the straight line in the low-frequency region indicated high ionic accessibility, whereas the arc with a small diameter in the high-frequency region revealed a small internal resistance of the cell. At a high current density of 1 A g$^{-1}$, the chrono potentiometry (CP) curve (Fig. 4b) exhibited a symmetrical triangular shape and a short charge–discharge time, showing a high reversibility of the supercapacitors. Both the

EIS and CP tests showed that the PCNF–50% PVA had the best electrochemical performance. The supercapacitors exhibited high rate capability and cyclic voltammogram (CV, Fig. 4c) curves did not deform when increasing the scan rates from 5 to 500 mV s$^{-1}$. In addition, we also plotted the charge–discharge curves of the supercapacitors at 1–10 A g$^{-1}$ to further verify the high rate capability (Supplementary Fig. 20). Gravimetric capacitances were calculated based on CV. At the scan rate of 5 mV s$^{-1}$, the capacitance was 163.6 F g$^{-1}$. The capacitance retained 132.06 F g$^{-1}$ at a high rate of 10 mV s$^{-1}$. With increasing the rate to 50 mV s$^{-1}$, the supercapacitor delivered a reasonable capacitance of 78.9 F g$^{-1}$ with a high capacitance retention of 88.5% after 9000 cycles (Fig. 4d). It is worth noting that the supercapacitors exhibited an elevated potential of 3.5 V, which was ~ 1 V higher than conventional ones, thereby greatly enhancing the energy density. For example, at 1 A g$^{-1}$, the supercapacitors delivered a capacity of 25.2 F g$^{-1}$ with a high power density of 1.75 kw kg$^{-1}$, whereas the energy density reached 42.77 Wh kg$^{-1}$. These values were notably higher than those of the previously reported CNFs and other porous carbon materials (Supplementary Table 2).

To further evaluate the excellent electrochemical performance of the PCNF–50% PVA, we fabricated sulfur cathodes with the PCNF films as scaffolds. The sulfur content in the whole cathode system was ~50 wt.%. Li-sulfur batteries were considered to be one of the most promising candidates for next-generation batteries due to their high energy density, but they suffered rapid capacity fade and low power density[43]. To make a Li-sulfur battery viable, a well-designed carbon network that affords an effective matrix for fast mass transport is the essential component. We postulate that the PCNFs work as open structures for maximizing both sulfur loading and optimal ion diffusion and electronic conduction.

The battery exhibited typical discharge voltage plateaus in the ranges 2.35–2.3 V and 2.05–2.02 V, and two charge voltage plateaus in the ranges 2.21–2.25 V and 2.35–2.39 V, at 1 C (Fig. 4e). The voltage plateaus in the 300th cycle coincided well with the initial cycle, indicating high reversibility of sulfur reactions in the cell. At an ultrahigh rate of 5 C, the voltage difference between the charge and discharge plateaus enlarged only ~0.03 V in comparison with the one at 1 C, suggesting fast sulfur reaction kinetics and high stability of the cathodes. This situation was consistent with the CV results (Fig. 4f), a slight shift in the oxidation peak to low value from the second cycle indicated the weakening of shuttle effects. At a high rate of 1 C, the batteries delivered high discharge capacities (Fig. 4g). The initial discharge capacity was 1380 mA h g$^{-1}$, corresponding to 82.3% of the theoretical capacity. After 300 cycles, the discharge capacity decreased to 1000 mA h g$^{-1}$, corresponding to 72.5% of the capacity retention. To investigate the effectiveness of the PCNF scaffolds, SEM was checked for the cathodes cycled up to 50 times at 1 C, as shown in Fig. 4h, i. It can be seen that the fresh cathode contained numerous sulfur agglomerates; however, after the 50th charging, the surface became smooth, suggesting that the PCNF network facilitated sulfur precipitation and avoided sulfur agglomeration into large particles.

## Discussion

PCNFs have many unique properties such as high surface areas, good mass diffusion, and superior accessibility to active sites. However, designing continuous CNFs with both high pore volume and large conductivity was an open problem. In particular, it is a great challenge to fabricate PCNFs with continuous and interconnected macropores. In general, the graphitic CNFs possessed high conductivity but limited pore volumes, whereas PCNFs with amorphous structures had reverse properties[44]. It was recently

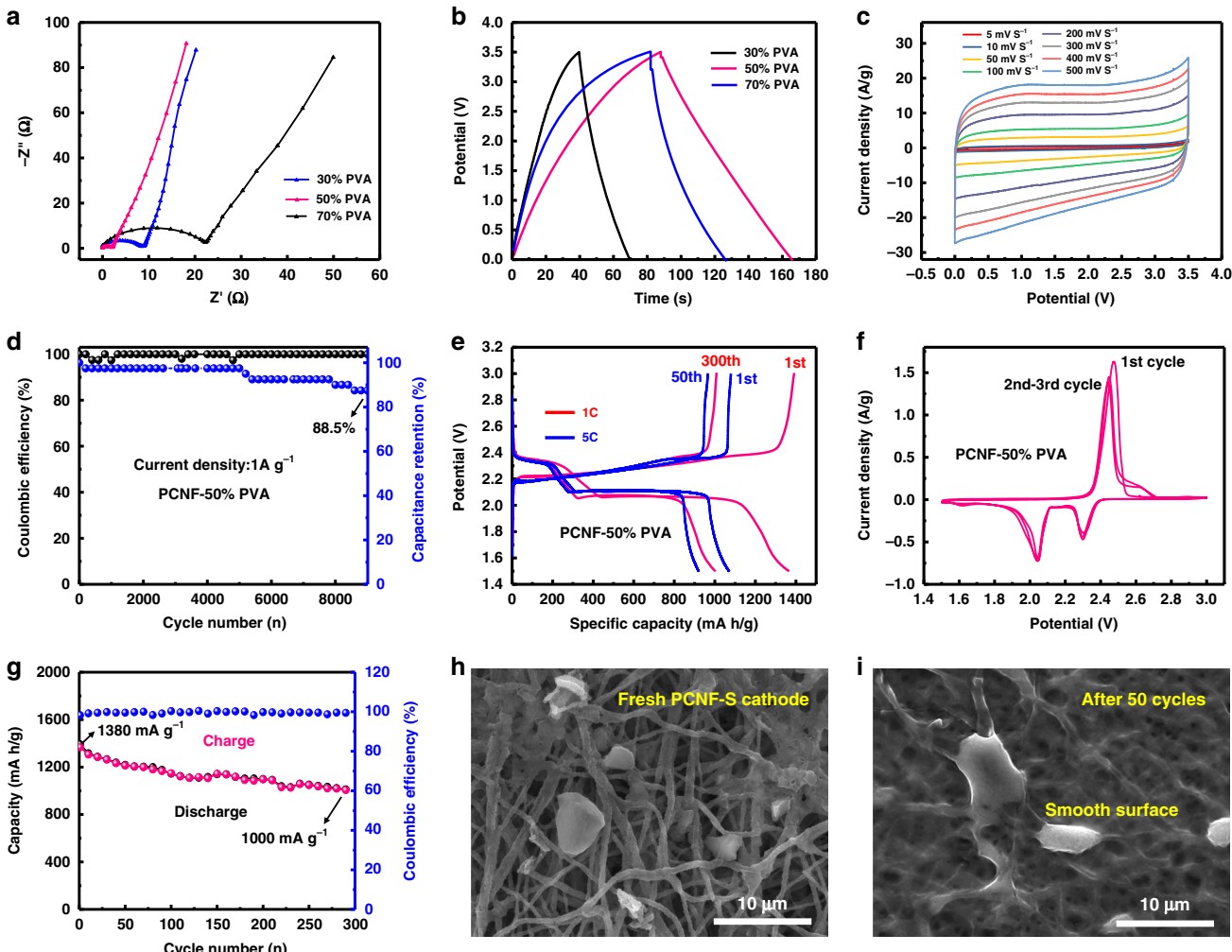

**Fig. 4 Electrochemical performance of the PCNF films. a–d** The performance of the freestanding PCNF electrodes in supercapacitors with Li-ion electrolytes. **a** Impedance spectra of the PCNF electrodes before cycling. **b** CP curves at a current density of 1 A g$^{-1}$. **c** CV curves of at four different scan rates. **d** Long cycling performance at 1 A g$^{-1}$. **e–i** The performance of PCNF-S cathodes in Li-S batteries. **e** Galvanostatic voltage profiles at 1 C and 5 C. **f** Continuous CV curves at a scan rate of 0.1 mV s$^{-1}$. **g** Long cycling performance at 1 C. **h**, **i** Surface morphology of the fresh electrode and the electrode after 50 cycles at 1 C, respectively. Source data are provided as a Source Data file.

reported that using a microphase separation method with block copolymers of PAN-polymethyl methacrylate as carbon precursors could produce PCNFs with uniformly distributed meso and micro pores[45]. However, preparing the block copolymers was time-consuming and needed to use a large amount of toxic organic solvents. In addition, such PCNFs exhibited low conductivity and only contained meso and micro pores with small pore volumes.

In this contribution, we enhanced the pore volume and electronic conductivity of PCNFs simultaneously by developing a facile and green method that combined both the macrophase and microphase separations. The solvent used for dissolving the precursor materials was water, which was very valuable for fabricating CNFs. To the best of our knowledge, the porosity of >80% is currently at the leading level, whereas the conductivity of 981 S cm$^{-1}$ was comparable with the graphene films[46–48]. It is noteworthy that two carbon plates (Supplementary Fig. 21) were adopted to sandwich the oxidized fibrous films during the high-temperature pyrolysis. The applied tension limited the agglomeration of PVA so that they did not shrink under tensile stress, whereas the porous structure provided a space for stress release, thereby stabilizing the PCNF films, which demonstrated excellent shape-memory performance and high flexibility.

To make the continuous and interconnected macropores in the nanoscale carbon fibers, the most important step was preparing the stable 3-D PVA-BA-PTFE sol networks, in which the PVA and PTFE were supramolecular assembled with BA as the cross-linking agent. The mass ratios of PVA and PTFE greatly affected the morphology of the PCNFs, as these two polymers could produce many different intermolecular interactions, which were important interfering factors for electrospinning, oxidation, and pyrolysis. For example, if the PVA content was too high, the as-spun fibers would stick together during the oxidation process and thus destroyed the fibrous structures (Supplementary Fig. 22). However, if the PTFE content was too high, the final PCNFs had lower surface areas due to the large pore sizes and small fiber diameters. Therefore, in our methods, without using activation chemicals, simply adjusting the mass ratios of PVA and PTFE allowed for self-regulation of the pore sizes and fiber diameters.

On the other hand, the conductivity of PCNFs was influenced by the pyrolysis temperature. A high pyrolysis temperature could improve the proportion of graphitic carbon, which enhanced the conductivity. In addition, we found the pyrolysis atmosphere also affected the conductivity. Without infiltrating dopants, the pyrolysis of the as-spun fibers in N$_2$ atmosphere directly yielded B-, F-, and N-doped PCNFs. As a comparison, with Ar as the

pyrolysis atmosphere, the PCNFs demonstrated lower conductivity (Supplementary Fig. 23) and there were not N in the PCNFs. Of note, the morphology of these PCNFs fabricated with both atmosphere had the similar porosity and morphology (Supplementary Fig. 24). Therefore, the N was probably caused by the reactions between $N_2$, BA, and F during the high-temperature pyrolysis[49]. A high temperature could accelerate the reaction. The required energy to initiate the reactions was shown in the Supplementary Information. These elemental doping greatly enhanced the conductivity[50]. The PCNFs with uniform macro pore structures and triply doped layered graphite sheets acted like macroscopic carbon nanotubes and could work as the wire to light the bulb without reducing the brightness (Supplementary Fig. 25).

To the best of our knowledge, this is the first report of fabricating sponge-like PCNFs with continuous macropores and large electronic conductivity. The successful synthesis of the PCNFs with interconnected macro, meso, and micro pores provided a reliable method to explore the autonomous control of porous structures and the improvement of electronic conductivity. Moreover, the cross-connection and entanglement between the CNFs could be automatically assembled into a 3-D nanofibrous film without using binders, which reduced the mass transfer resistances. In particular, the thickness, packing density, and the size of the PCNF films were controllable by simply changing the electrospinning time and voltage, and designing coupled electrospinning devices. These features promised the applications of the freestanding PCNF films in high-power and -energy density supercapacitors and batteries, which required both large conductivity and high porosity.

In summary, we have reported a chemical crosslinking electrospinning, an economical and environmentally friendly method for the scalable fabrication of PCNFs with ultrahigh porosity and outstanding conductivity. Both the macrophase separation of PTFE NPs and the microphase separation of PVA and BA contributed to the formation of highly controlled trimodal porous structures that contained macro, meso, and micro pores. The freestanding PCNF films exhibited integrated characteristics of porous nanostructure, high surface areas of $>750\ m^2\ g^{-1}$, large conductivity of $>980\ S\ cm^{-1}$, and excellent mechanical properties of non-brittle cracking after deformations. Importantly, the flexible PCNF films demonstrated excellent functions in many fields such as environmental and energy. The versatility of the method solved the defects of non-expansion or high cost in the conventional preparation of PCNFs and enabled the development of advanced applications beyond the reported applications in this paper, such as electro-catalytic carriers, filters, and wearable electronics.

## Methods

**Fabrications of sponge-like PCNFs**. To make the stable PVA-BA-PTFE sol, we first prepared a 5 wt.% of BA solution by dissolving BA powders (with a melting point of 184 °C) in the deionized water. Then, 15 wt.% of PVA solution was prepared by dissolving PVA powders (PVA 1788, Mw 83,000–85,000, Aladdin) in the deionized water at 90 °C for 3 h with a constant stirring rate. After cooling, the BA solution was added into the PVA solution and the mixture was stirred for 1 h. The BA amount was controlled within 3 $\mu L\ g^{-1}$ of PVA. After that, the PTFE (120 nm) water emulsion (60 wt.% of solid content) was put in the above mixture solution and was stirred for 3 h. The mass ratios of the PVA and PTFE were adjustable. The electrospinning process was carried out with a feed rate of 1.5 $mL\ h^{-1}$, a voltage of 22 kV, and a distance of 18 cm between the needle and the drum collector. The humidity and temperature of the electrospinning chamber were 45 ± 5% and 25 ± 2 °C, respectively. After electrospinning, the as-spun fibers were kept in a vacuum oven at 280 °C for 3 h and then were calcined at 800~1200 °C in $N_2$ or Ar for 2 h with a heating rate of 5 °C $min^{-1}$ to obtain PCNF films.

**Material characterizations**. Morphology and elemental distribution of the samples were examined by SEM (Hitachi S-4800) and TEM (JEM-2100F). Crystal structure characterization was investigated using Bruker XRD with Cu Kα radiation

between 10° and 90°. The graphitization degrees and the defect characteristics of the PCNFs were checked by Raman spectrometer with an excitation wavelength of 532 nm. The molecular chain structures were analyzed by FTIR (Nicolet iS10). The surface areas, pore volumes, and pore size distribution of the PCNFs were measured with Brunauer–Emmett–Teller analyzer (ASAP 2460, Micromeritics, Co. USA). The thermal decomposition process were determined by TGA (SDT Q600) in $N_2$ atmosphere at the heating rate of 5 °C $min^{-1}$ from room temperature to 800 °C and the $CO_2$ adsorption properties of the samples were examined at 25 °C in $CO_2$ atmosphere. The conductivity of the PCNF films was tested by a four-probe tester (ST-2258C) according to the standard of JJG508-87. The porosity was determined with a weighing method by infiltrating the film in tert- butanol solution. The UV-Vis absorption spectra was performed by U-3900 Hitachi spectrophotometer.

**Electrochemical measurements**. *Supercapacitors*. CR2025-type coin cells were used as the testing PCNF based all-carbon symmetric supercapacitors. Two freestanding circular PCNF films with equal sizes (diameter of 1.25 cm) were used as the counterpart electrodes, the Cellgard 2400 were used as separators, and 1 M $LiPF_6$ in ethylene carbonate, diethyl carbonate, and ethyl methyl carbonate (EC : DEC : EMC = 1 : 1 : 1 v/v) were used as electrolytes. CV and EIS were performed on the same electrochemical workstation (Chenhua, CHI 660E, Shanghai). The potential window and the scan rates of the CV were 0~3.5 V and 5~500 mV $s^{-1}$, respectively. EIS experiments were conducted in a frequency range from 100 kHz to 0.1 Hz at a potentiostatic signal amplitude of 10 mV. Constant current charge and discharge tests were carried out at a current density of 1 A $g^{-1}$ ~10 A $g^{-1}$ and within 0~3.5 V. All experiments were kept at room temperature.

*Li-S batteries*. CR2025-type coin cells were used as the testing cells. Ring Li-foil with a thickness of 20 μm and diameter of 1.65 cm was used as the anode, Cellgard 2400 microporous membranes as separators, 1.0 mol $L^{-1}$ bis(trifluoromethane sulfonyl)imide/bis(fluorosulfonyl)imide (1:1 mass ratio) and 0.1 mol $L^{-1}$ $LiNO_3$ dissolved in 1,2-dimethoxyethane and dioxolane (1:1, v/v) as electrolytes. The sulfur content was 50 wt.%. Electrochemical measurements were performed galvanostatically between 1.5 and 3.0 V at various current densities. Capacities were calculated based on the weight of sulfur. CV experiments were conducted at 0.1 mV $s^{-1}$. EIS measurements were carried out in a frequency range between 100 kHz and 0.1 Hz at a potentiostatic signal amplitude of 5 mV. All experiments were conducted at room temperature.

## Data availability

All the experimental data that support the findings of this study are available from the corresponding author upon reasonable request. The raw data underlying Figs. 2f, g, h, 3a, c, d, e, f, and 4, as well as Supplementary Figs. 5, 6, 7, 8, 13, 16, and 20 are provided as a Source Data file.

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

## Acknowledgements

This work is supported by the National Natural Science Foundation of China (No. 51702047 and 51873029), the State Administration of Science, Technology, and Industry for National Defense, PRC (JCKY2018203c035), the National Key R&D Program of China (No. SQ2018YFC200227), Program of Shanghai Academic Research Leader (No.18XD1400200), Innovation Program of Shanghai Municipal Education Commission (No. 2017-01-07-00-03-E00024), and Young Elite Scientists Sponsorship Program by CAST (No. 2018QNRC001).

## Author contributions

J. Yan and B.D. conceived and designed the project. J.Yan and K.D. conducted the synthesis and applications of PCNFs, and wrote this paper. K.D., Y.Z., X.W., and A.A. conducted material characterizations. J. Yu helped to analyze experimental results. All authors contributed to discussing and revising the paper.

## Competing interests

The authors declare no competing interests.
