## [Peer Review File · Nature Communications]

Reviewers' comments:

Reviewer #1 (Remarks to the Author):

This manuscript demonstrates a chemical crosslinking electrospinning, a macro-micro dual phase separation method to synthesize sponge-like porous carbon nanofibers (PCNFs), which have developed porous architecture and high conductivity. The flexible membranes have multifunction and many applications, such as adsorption and energy storage. The structure of the sponge-like porous carbon nanofibers is interesting. However, the reviewer can not be recommend the publish of this manuscript on Nature Communications as some major results are inconsistent and inaccurate.

(1) "The PCNFs were doped with B, F and N (Fig. 1f)". XPS is needed to characterize the content of B, F and N in the PCNFs. "The N was from the N₂ atmosphere during the process of high temperature". Most of carbon materials are prepared under N₂ atmosphere, but few are nitrogen-doped without additional nitrogen sources. Please give explanations.

(2) To characterize the pore properties of PCNFs, only N₂ adsorption/desorption measurements is not adequate. Especially the materials have lots of macropores, for which N₂ sorption is unavailable. The porous structure needs to be characterized not only by N₂ sorption but also using mercury porosimetry measurement. "Figure S12. N₂ adsorption-desorption isotherms of the different PCNFs that produced with different PVA contents". However, the given curves are pore size distribution curves, not N₂ adsorption-desorption isotherms.

(3) The influence of the mass ratio of PVA and PTFE on the porosity parameters including surface area, pore sizes and pore volume, and the reason should be discussed.

(4) It is claimed that the obtained PCNFs has a high graphitization degree. However, all the characterization including XRD, Raman and HRTEM cannot support this conclusion. Figure S6 shows the XRD pattern of PCNF-1200 °C has a wide peak at 25°, indicating a typical amorphous carbon. Although a few graphitic ribbons can be observed from TEM in Figure S11, the average inter-planar distance of ~0.41nm cannot match well the d-spacing of (002) in graphitic carbon (the d₀₀₂ of graphite is 0.334), characterizing typical amorphous carbon.

(5) How to understand so high conductivity of the PCNF? It is amorphous carbon, but the conductivity of 980 S/cm is "two to three orders of magnitude higher than normal CNFs", and even much larger than graphene. The metallic conductivity of the film (980 S/cm) was ascribed to the B, F and N triply-doping in the material, which is not convincing. Why and how doping enable it have so high conductivity?

(6) The film has interconnected macro-meso-micro porous structure. Discussion on the pore creation mechanism and how to control is needed.

(7) This authors used a chemical crosslinking electrospinning, a macro-micro dual phase separation method to prepare the sponge-like porous carbon nanofibers and described it as a low-cost scalable approach. What is the electrospinning speed?

(8) For many applications, mechanical strength is an important parameter. Please give the mechanical performances of the membrane.

(9) The PCNF doped with B, F and N is hydrophobic with a contact angle as high as 140.5°. Why?

(10) The adsorption of PCNFs to ethanol DME, DOL are all 50 times of the weight of itself. What's the adsorption mechanism of PCNFs to these liquids? Is there any correlation between the adsorption capability and the molecular structure?

(11) The author called the as-prepared all-carbon symmetric supercapacitors with LiPF₆ as electrolyte as "Li-ion supercapacitors". This is not correct. Li-ion supercapacitors are asymmetrical capacitor using lithium storage materials as one electrode. The capacitor assembled in this work is a typical double layer capacitors, not Li-ion supercapacitors.

(12) When used as supercapacitor electrode, there are some errors in the electrochemical performance test of the film. In the impedance spectra of the film, the Y-axis associated with the imaginary part of the measured impedance should be negative values. The chrono potentiometry curves should be tested in the potential range of 0 - 3.5 V instead of -3.5V - 3.5V. The CV curves are not rectangular, and the peaks at high potential means that the materials cannot endure so high potential, which can be confirmed by the low columbic efficiency. Furthermore, for supercapacitor electrode, 200 cycles is

not long enough to evaluate the cycle stability. How was the high energy density of 93.65 Wh/Kg calculated? What's the capacitance by charge-discharge?

(13)As scaffolds for sulfur loading in Li-S battery cathode, it can only accommodate sulfur content of ~50 wt.%. It is not an attractive value, far from maximizing S loading. Besides, it can be seen that the columbic efficiency is not very good. So why the capacity increase gradually from 5 to 50 cycles? Longer cycle performance measurement should be supplied.

(14)In Figure 5, the scale of the weight loss need be given.

(15)What's the yield of the fiber?

(16)There are still some spelling errors.

Reviewer #2 (Remarks to the Author):

The analysis needs additional clarification as described below, 1) The study on the synthesis of a flexible, porous, and electrically conducting carbon foam via a strategy of combining microphase and macrophases separations is interesting. In comparison with the previous reported porous carbon, what's the striking advance in fundamental understanding your carbon structures and what are the merits of the strategy in preparing the materials? In order to prove the advancement of this study, I recommend you to refer more papers published in recent years.

2) The second point is about the flexibility of the porous carbon nanofibers. The authors stated that the pre-oxidation process could enhance the stability of the primary fibers, but the explanation of the phenomenon is not clear, you need to add more discussions. In addition to demonstrate the flexibility of the film by using some digital figures, you also need to test the flexible properties like stress-strain. Last, you should compare the flexibility of your materials with the reported ones, since it's amazing to maintain well of the porous structures.

3) The third important point is about the conductivity of the porous carbon nanofibers. Is the conductivity from a single nanofiber or from the whole film? The resistance of such flexible (or freestanding) film will increase once its area is increased. You should test at least two different samples with different areas. You also should pay more attention to the root cause of the high conductivity.

4) The fourth important point is the wide applications of the porous carbon nanofibers. Firstly, the porous carbon membranes exhibited both hydrophobicity and rapid electrolyte wettability, can the authors explain the mechanisms of these two different phenomena? The insert figures in Figure 3d does not show the phenomenon you described and I recommend to re-edit this image. Secondly, it is interesting that the porous carbon materials had such a wide range of applications. Which of these applications do you think is more likely to be commercialized? I recommend to highlight its advantages and give more descriptions.

5) Overall, this paper needs some refinements and more focus on the details about the mechanisms of porosity and high conductivity. It is also important to state over whether these structures can be reproducibly created and what's the production scale?

Reviewer #3 (Remarks to the Author):

The manuscript "Multifunctional Flexible Membranes from Sponge-Like Porous Carbon Nanofibers with Metallic Conductivity" by Dong et al. reports on a novel approach to fibrous carbon sponges with electric conductivity by electrospinning, followed by pyrolysis.

The manuscript is surely of interest for the readers of Nature Communication but requires major revision prior to publication due to some flaws and unclear information. It also appears that some more in depth analysis but somewhat less on application would have been better.

A general remark: page numbers and line numbers would make the review and revision process easier.

The reported electrical conductivity is by two orders of magnitude below the conductivity of metals. Therefore, the term "metal-like" conductivity cannot be applied.

Refer to the sentence: "The concept of freestanding electrode has been reported in a number of papers." Papers should be cited here. A simple Google search on sponges or electrospinning and electric conductivity brings many papers for porous scaffolds including carbons with high conductivity and even higher, truly metal-like conductivity. The authors should cite these papers and discuss the novelty of their work in the light of relevant literature.

Refer to the sentence: "At this temperature, the semi-fluid PVA had a..." Semi-fluid is too qualitative for any reproduction. Please give quantitative data for the viscosity of the formulation.

Refer to the sentence: "The PCNFs were doped with B, F and N (Fig. 1f)." Here, sound elemental analysis is missing, e. g. by EDX. Otherwise this remains mere speculation.

Fig. 1. The image of the fibrous sponge-like fiber is too speculative. Clearly, a cross-sectional SEM image is missing in order to verify that pores are really continuously.

Fig. 2e. Nice demonstration of robustness but here quantitative mechanical data are absolutely essential.

Fig. 3a. at present too speculative – see above

Point-by-point responses to Reviewers' critiques

Note: All critiques are highly appreciated. The changes are **highlighted in blue** in the **revised Manuscript** and **revised Supplementary Information** for review as suggested.

Comments from reviewers:

Reviewer #1:

Overall comment: This manuscript demonstrates a chemical crosslinking electrospinning, a macro-micro dual phase separation method to synthesize sponge-like porous carbon nanofibers (PCNFs), which have developed porous architecture and high conductivity. The flexible membranes have multifunction and many applications, such as adsorption and energy storage. The structure of the sponge-like porous carbon nanofibers is interesting. However, the reviewer cannot be recommended the publish of this manuscript on Nature Communications as some major results are inconsistent and inaccurate.

Response: *We thank the reviewer for the valuable comments on our manuscript; the comments are very helpful in improving our manuscript. On the basis of the sincere suggestions, we have appropriately revised the manuscript and added additional experiments and analysis clarification to make the paper easier to follow and to clearly show our main achievements. We hope that the reviewer finds our responses satisfactory and convincing. The responses to comments are listed below:*

Point-by-point response

Comment 1-“The PCNFs were doped with B, F and N (Fig. 1f)”. XPS is needed to characterize the content of B, F and N in the PCNFs. “The N was from the N₂ atmosphere during the process of high temperature”. Most of carbon materials are prepared under N₂ atmosphere, but few are nitrogen-doped without additional nitrogen sources. Please give explanations.

Response: *Thank the reviewer for the sincere comments. We fully understand the concerns about the N-doping since it's difficult to break the nitrogen-nitrogen triple bond of N₂. However, the N was indeed found in the PCNFs that fabricated under the pyrolysis atmosphere of N₂. The EDS (**Figure 1f**) was tested with a very low speed along with the TEM figure, in which the distribution of N coincided well with the porous structures in the single PCNF, may eliminating the possibility of N₂-adsorbing*

into the pores of the PCNFs.

To further confirm this, and combined with the sincere suggestion from the Reviewer of #3, we added a series of experiments in the revised manuscript. First, we compared the conductivity of the PCNFs that produced under different gas atmospheres of Ar and N₂ at three different pyrolysis temperatures (800, 900 and 1000 °C) (**Figure S22**). Although there were no obvious difference on the morphology and porosity (**Figure S23**), the PCNFs that fabricated under Ar always exhibited lower conductivity than the PCNFs that created under N₂. This result indicated that the N₂ atmosphere had great effect on the conductivity of the PCNF film. We used a high energy EDS mapping to retest the PCNF film, there was obviously lots of N-atoms, as shown in the following figure. In addition to the EDS, we also tested XPS in the revised manuscript, as shown in **Figure S8**. The atomic percents of C, O, N, B and F were 94.33%, 3.15%, 1.25%, 0.93% and 0.34%, respectively. After checking some books and papers (ex. *J. Am. Chem. Soc.* 2019, **141**, 2884-2888; *J. Chem. Eng.* 1963, **8**, 1), the N-doping was probably caused by the reactions between N₂, BA and F during the high temperature pyrolysis. According to the Gibbs function, these materials could react at room temperature. We have added these discussions in the revised manuscript (**Lines 392-403, and Page 2 of the Supplementary Information**).

Comment 2-To characterize the pore properties of PCNFs, only N₂ adsorption/desorption measurements is not adequate. Especially the materials have lots of macropores, for which N₂ sorption is unavailable. The porous structure needs to be characterized not only by N₂ sorption but also using mercury porosimetry measurement. “Figure S12. N₂ adsorption-desorption isotherms of the different PCNFs that produced with different PVA contents”. However, the given curves are pore size distribution curves, not N₂ adsorption-desorption isotherms.

Response: *We highly appreciate the reviewer for these appreciated questions. We agree with the reviewer that the N₂ adsorption-desorption measurements could not accurately determine the properties of macro pores. We have added more discussions in the revised manuscript (lines 185-188). To determine the sizes of these macro pores, we used both the BET-measurement and the TEM characterization. According to reference 28, using BET could roughly measured the size of the macro pores. When P/P₀ was > 0.9, the N₂ absorption increased sharply, which indicated that there were large amounts of macro pores in the PCNFs. The pore size was measured as around 68 nm, which was consistent with the roughly measured value by TEM. While the pore volume was determined with the weighing method by immersing the PCNF films into tert-butanol. Actually, we had tried at least three times to characterize the macro pores by using mercury porosimetry measurement, but the results (~1.5 um) were not good due to the breaks of the film during the test. When using this method, the exerted pressure was more than 200 MPa. However, the mechanical strength of the PCNF film was 2.23 MPa although the film had a large Young's modulus of 429 MPa (we have added these data into the revised manuscript, as shown in **Figure 3a** and **3b**). The film could not stand such large pressure, and it cracked during the testing period. For the second question, we have revised it (**Figure S16**) as "pore distributions of the different PCNFs that produced with different PVA contents".*

Comment 3-The influence of the mass ratio of PVA and PTFE on the porosity parameters including surface area, pore sizes and pore volume, and the reason should be discussed.

Response: *Thanks for this appreciated question. In our experiments, the mass ratio of PVA and PTFE had great effects on the morphology of PCNFs. To clarify these effects, we added more experiments and added more discussions in the revised manuscript (lines 119-124; lines 131-134; lines 146-151; lines 154-156; lines 383-385; **Figure S2, S9** and **S14**). First, without using PTFE, the as-spun PVA nanofibers could not transform into carbon nanofibers after pyrolysis, since PVA started to melt at ~150 °C and flowed into a film at ~200 °C (**Figure S2**). Actually, once the content of PVA was larger than 80 wt.% in the PVA-PTFE system, the final products were not fibers, but like flat films (**Figure S10** and **S21**). Correspondingly, the porosity decreased a lot, since the macro pores were mainly from the decomposition of PTFE. PTFE could prevent the diffusion of PVA at a higher temperature, ex. 280 °C. At this temperature, the chains of the PVA changed into stable conjugated C=C bonds, enhancing the*

*stability of the final PCNFs. We mainly compared three different ratios of 7:3, 5:5 and 3:7 (PVA:PTFE), and the detailed specific surface areas and the pore volumes of the PCNFs that produced with different PVA contents were summarized in **Table S1**. The mechanisms of forming the uniformly distributed pores were summarized as: the micro and macro phase separations brought a homogeneous pore distribution, in which small molecule (BA and PVA) pyrolysis created meso and micro pores, while the decomposition of large PTFE NPs left continuous macro pores.*

Comment 4-It is claimed that the obtained PCNFs has a high graphitization degree. However, all the characterization including XRD, Raman and HRTEM cannot support this conclusion. Figure S6 shows the XRD pattern of PCNF-1200 °C has a wide peak at 25°, indicating a typical amorphous carbon. Although a few graphitic ribbons can be observed from TEM in Figure S11, the average inter-planar distance of ~0.41nm cannot match well the d-spacing of (002) in graphitic carbon (the d002 of graphite is 0.334), characterizing typical amorphous carbon.

Response: *Many thanks for this respected comment, and we truly appreciate this critical question. We have revised the manuscript carefully and changed the word “graphitic” as “a mixture graphitic, turbostratic and amorphous” (line 138; lines 172-175, Figure 2d and S15). According to the TEM figures, we retested the d-spacing. The average d-lattice was around 0.41 nm. Generally, the highly graphitized carbon lattice is around 0.34 nm. However, in our work, the CNFs had a mixture of graphitic, turbostratic and amorphous carbon domains with different ratios depending on the degree of graphitization, which was mainly influenced by the carbonization temperature as reported earlier (Adv. Energy Mater. 2016, 6, 1501588 and Adv. Energy Mater. 2013, 1301448). According to these literature, to get highly or fully graphitized carbon with d-spacing around 0.34 nm, it required very high carbonization temperature around 3000 °C, which was away from ours as we only used 800~1200 °C for only 2 h. In addition, researchers also reported the similar d-spacing of 0.41 nm for graphitic carbons previously (Nanoscale, 2012, 4, 6800–6805; Energy Environ. Sci., 2014, 7, 2689-2696; et al.), which may also support our results.*

Comment 5-How to understand so high conductivity of the PCNF? It is amorphous carbon, but the conductivity of 980 S/cm is “two to three orders of magnitude higher than normal CNFs”, and even much larger than graphene. The metallic conductivity

of the film (980 S/cm) was ascribed to the B, F and N triply-doping in the material, which is not convincing. Why and how doping enable it have so high conductivity?

Response: *Thanks for these appreciated questions. We fully understand the reviewer's concerns about the conductivity. We removed the word of "metallic conductivity", and added more discussions about the conductivity in the revised manuscript (Lines 390-399). Combined with the sincerely comments from the reviewer #3, we would like to illustrate these comments from three aspects.*

First, the high conductivity of 980 S/cm was achieved from the PCNFs that produced at 1200 °C. As stated in the comment 4, such a high temperature rendered the PCNFs with a mixture of graphitic, turbostratic and amorphous structures. However, with a low calcination temperature such as 800 °C, the conductivity was only 700 S/cm. There were some reports on CNFs or CNF/graphene composite fibers that produced at a low temperature but exhibited very high conductivity of larger than 1000 S/cm (e.g. Carbon 2016, 99, 407-415; Nanoscale 2013, 5, 1183-1187; ACS Appl. Mater. Interfaces 2019, doi.org/10.1021/acsami.9b08198; Carbon 2010, 48, 4421-4431; Adv. Energy Mater. 2018, 8, 1801854; J. Mater. Chem. C 2015, 3, 6589-6599; et al.), some of them reached 12000 S/cm

Second, from the cross-sectional SEM (Figure S14) and TEM (Figure 2c) images, our PCNFs had thorough porous structures. That means, each single CNF was composed by numerous thinner carbon sticks with only several nanometers. In our previously report (Nat. Commun. 2019, 10, 1458), we verified that such kind of carbon nanofiber (with a diameter smaller than 30 nm) membranes had superior electrical conductivity of ~ 750 S/cm. Of course, the deeply analysis for the conductivity enhancement need to be clarified in the future, and we are investigating the conductivity mechanism of such kind of PCNFs.

Thirdly, unlike the conventional synthesis that rely on blending pitch or PAN with the doping elemental materials, we adopted BA, PVA and PTFE to fabricate PCNFs. Here, there were already a lot of C-F bonds in PTFE, while the crosslinking reactions between PVA and PTFE with BA as the crosslinking agent led to stable C-B bonds. We think there maybe some new mechanisms for forming C-N bonds in the PVA-PTFE based PCNFs under N₂ atmosphere. In the revised manuscript, we compared the conductivities of PCNFs that were prepared by varying the gas atmosphere (Figure S22 and S23). Although the N₂ atmosphere could produce higher conductivity, the conductivity of the PCNFs that produced with Ar atmosphere were still much larger than the traditional CNFs that produced without using PTFE (eg. iScience 2019, 16,

122-132), indicating the C-F and C-B had great effects for improving the conductivity. The covalently functional groups of C-B, C-N and C-F increased the contents of SP² structures, rendering the CNFs with a high conductivity. A lot of published work verified the effects of B, N and F doping on enhancing the conductivity of carbon (Prog. Mater. Sci. 2013, 58 565–635; Solid State Sci. 2011, 13 1459e1464; Nano-micro Lett. 2019, 11, 9), the conductivity of such carbons could also reach several hundred S/cm.

Comment 6-The film has interconnected macro-meso-micro porous structure. Discussion on the pore creation mechanism and how to control is needed.

Response: Many thanks for this respected comment. We have added more discussions in the revised manuscript (Lines 131-134; Lines 154-156; Lines 383-385; Figure S6, Figure S9, Figure S10, Figure S12, Figure S14, Figure S21, Figure S23). According to our results, both the micro and macro phase separations brought the homogeneous pore distribution. Small molecule (BA and part of PVA) pyrolysis created meso and micro pores, while the decomposition of large PTFE NPs (~120 nm size) left continuous macro pores. Since the PTFE itself would transformed into carbon after pyrolysis, the size of the macro pores was much smaller than 120 nm. Therefore, to control the pore size and pore distribution, the distributions of PVA, BA and PTFE in the as-spun nanofibers needed to be optimized. First, the mass ratios of these precursor materials were very important for controlling the porosity. The mass ratio could influence the stability and properties of the electrospinning sol, and thus would affect the morphology of the as-spun nanofibers after electrospinning. Second, the pre-oxidation process and the pyrolysis temperature was also essential for controlling the porosity. PTFE could stable the PVA and facilitated the formation of stable conjugated C=C bonds in PVA during the pre-oxidation. If skipping the pre-oxidation process, we found that the as-spun PVA-PTFE fibers could not transform into PCNFs (Lines 119-124; Lines 128-129). While the high temperature pyrolysis also had great influence on the morphology of PCNFs. The higher temperature always led to thinner PCNFs and higher porosity.

Comment 7-This authors used a chemical crosslinking electrospinning, a macro-micro dual phase separation method to prepare the sponge-like porous carbon nanofibers and described it as a low-cost scalable approach. What is the electrospinning speed?

Response: Thanks for this appreciated question. In our experiments, 20 syringes were

used in parallel for electrospinning. For each syringe, the electrospinning speed was set as 1.5 mL/h. With respect to the large-area fabrication, even industrial process, two important issues should be carefully considered: solvents and fabrication process. Here, the solvent was deionized water. However, to make the PAN-based CNFs, the solvent was always toxic and expensive DMF. In addition, because of the simplicity of the self-assembly process of our methodology and the facile availability of the designed precursor solutions, we think scaling up the synthesis of PCNFs was possible. Actually, in our previous study, using our lab equipment (DXES-V spinning machine, SOF Nanotechnology Co., Ltd., China), we could easily obtain continuous membranes with a large width of 120 cm, as shown in the following figures. Therefore, we have confidence in the fabrication of such PCNF membranes with larger areas using more syringes and larger collectors. (lines 413-415)

Comment 8-For many applications, mechanical strength is an important parameter. Please give the mechanical performances of the membrane.

Response: Many thanks for this respected comment. Combined with the reviewers of #2 and #3, we have added these data in the revised manuscript (**Figure 3a and 3b**), and removed the initial two figures. The corresponding illustrations were also added (**lines 222-229**). The PCNF films delivered a high Young's modulus of 429 MPa with a mechanical strength of 2.23 MPa and an extended strain of 0.52% before failure. In addition, the PCNFs exhibited a bending rigidity of only 13.1 mN, which was much soft than the value of paper towel. Such mechanical performance made it possible to apply these PCNFs into real applications for polluted gas adsorption, sewage disposal, liquid storage, energy storage and so on.

Comment 9-The PCNF doped with B, F and N is hydrophobic with a contact angle as high as 140.5°. Why?

Response: Thanks for this appreciated question. We have added more discussions in the revised manuscript (**Lines 257-263**). According to our understanding, there are two reasons.

First, water droplets are objects with clearly defined surface and surface tension. When they contact the surfaces of nano-size structure, the air will absorb the voids of the nano-grooves and forms "air cushion" between the droplets and the surface, which can effectively prevents the water droplets from infiltrating.

Second, superior hydrophobic surfaces can be prepared by two ways: one is to change the surface roughness and morphology of the materials; the other one is to modify low surface energy materials on the surface of the materials with a certain roughness. For one thing, the porous structure of the synthesized PCNFs increases the roughness of the surface, thus increasing the hydrophobic properties. For another, the F doping decreased the critical surface tension of the PCNF membrane. Specifically, the C-F group had a very low surface energy since the C-F bond had a high energy but a low polarization. F was regarded as an hydrophobic agent and the F-containing polymers were always applied for fabricating waterproof textiles (ref. 37, 38). Therefore, the super hydrophobic property could be achieved by combining the F-doping with the surface roughening. Silicone/fluorine materials are the most commonly used and important hydrophobic materials with low surface energy, especially the surface energy of perfluoroalkane was only 10 mN/m.

Comment 10-The adsorption of PCNFs to ethanol DME, DOL are all 50 times of the weight of itself. What's the adsorption mechanism of PCNFs to these liquids? Is there any correlation between the adsorption capability and the molecular structure?

Response: We thank the reviewer for these appreciated comments. According to our understanding from the perspective of textiles, there are four reasons.

First, the structures of carbon materials would change when they were doped by heteroatoms, and the carbon surface would take polar groups. There were many oxygen atoms and other atoms including B, N and F in the PVA-based PCNFs, these polar groups had strong interactions, such as Van der Waals force, with the organic solvents of DME, EG and DOL. **Second**, PCNFs are inorganic materials with large surface tension and high surface energy, which enhanced the wicking ability by organic liquids; **Thirdly**, the synthesized PCNFs have a large number of macro pores, meso pores and micro pores, which increased the surface roughness and enlarged the wetting spreading areas of the organic liquids. **Fourthly**, the electrospun PCNFs had

small diameter and high porosity between the CNFs, which rendered the film with a strong capillary adsorption effect and thus improved the liquid wettability and store capacity. We have added some more discussions in the revised manuscript (lines 282-285).

Comment 11-The author called the as-prepared all-carbon symmetric supercapacitors with LiPF₆ as electrolyte as “Li-ion supercapacitors”. This is not correct. Li-ion supercapacitors are asymmetrical capacitor using lithium storage materials as one electrode. The capacitor assembled in this work is a typical double layer capacitors, not Li-ion supercapacitors.

Response: *Thanks for the appreciated question. It is indeed a misunderstanding of the concept of “Li-ion supercapacitors”, we have corrected the phrase as “all-carbon symmetric supercapacitors” throughout the whole revised manuscript.*

Comment 12-When used as supercapacitor electrode, there are some errors in the electrochemical performance test of the film. In the impedance spectra of the film, the Y-axis associated with the imaginary part of the measured impedance should be negative values. The chrono potentiometry curves should be tested in the potential range of 0 - 3.5 V instead of -3.5V - 3.5V. The CV curves are not rectangular, and the peaks at high potential means that the materials cannot endure so high potential, which can be confirmed by the low columbic efficiency. Furthermore, for supercapacitor electrode, 200 cycles is not long enough to evaluate the cycle stability. How was the high energy density of 93.65 Wh/Kg calculated? What’s the capacitance by charge-discharge?

Response: *Many thanks for these appreciated questions. First of all, it’s our mistake to make wrong Y-axis coordinates, we have corrected them in the revised version (Figure 4a).*

Second, after checking some references (eg. Chem. Mater. 2012, 24, 433-443; Energy Storage Mater., 2016, 5, 103-110; J. Colloid Interface Sci., 2018, 527, 40-48; J. Power Sources, 2010, 195, 7120-7125), we found some researchers tested the double layer supercapacitors in the potential range of $-m V - m V$ ($m=0.5, 0.4, 1.8$ and 0.6), that’s why we set our potentials as $-3.5V - 3.5V$. We could achieve a high energy density under this wide electrochemical window. Of course, we fully understand the reviewer’s considerations and we have changed the voltage range as $0 - 3.5 V$ and re-characterized the performance of these supercapacitors in the revised

manuscript, as shown in **Figure 4b** and **4c**. After changing the testing potential window, at the scanning rates of 5, 10, 20 and 50 mV/s, the CV curves presented well. Thirdly, we retested the long cycling stability of the supercapacitors and increased the cycle number to 8000 in the revised manuscript, as shown in **Figure 4d**.

For the calculation of energy density, we used the following formula: $E = 0.5 * C * (\Delta V)^2$, where $C = It / (m \cdot \Delta V)$, t is the discharge time. After changing the testing voltage potential range, the new energy density was calculated as 42.77 Wh/Kg at 1A/g, and the new capacitance by charge-discharge was calculated as 163.6F/g under a scanning rate of 5 mV/s. We have revised all of these new data in the manuscript (**Lines 315-322**).

Comment 13-As scaffolds for sulfur loading in Li-S battery cathode, it can only accommodate sulfur content of ~50 wt.%. It is not an attractive value, far from maximizing S loading. Besides, it can be seen that the columbic efficiency is not very good. So why the capacity increase gradually from 5 to 50 cycles? Longer cycle performance measurement should be supplied.

Response: Many thanks for these appreciated comments. We agree with the reviewer that the ~50% wt.% of sulfur was not an attractive value in comparison of the reported high sulfur loading cathodes. However, we did not use current collector, which also accounted a large weight ratio in the whole battery system. At a high current rate of 1C, the battery needed time for activation, that's might the reason of the gradually increasing capacity from 5 to 50 cycles. To avoid such situations, in the revised manuscript, we re-assembled the batteries and activated the fresh batteries at 0.05 C for 2 cycles, and then started the round of cycles at 1 C. A longer cycle of 300 have been added in the new version (**Figure 4e** and **4g**).

Comment 14-In Figure 5, the scale of the weight loss need be given.

Response: Thanks for this appreciated question. We have revised the previous Figure S5 and marked the weight loss (**Figure S6**) in the revised supplementary information.

Comment 15-What's the yield of the fiber?

Response: Thanks for this appreciated question. We have added discussions on this question in the revised manuscript (**Lines 135-137**). According to the TG analysis, the total carbon yield was 4.75%.

Comment 16-There are still some spelling errors.

Response: *Thanks and we have revised the manuscript carefully and highlighted the revisions throughout the manuscript.*

Reviewer #2:

Overall comment:

The analysis needs additional clarification as described below. The study on the synthesis of a flexible, porous, and electrically conducting carbon foam via a strategy of combining micro-phase and macro-phases separations is interesting.

Response: *We thank for the reviewer's recognition on our work; the comments are very helpful in improving our manuscript. We have appropriately revised the manuscript and added additional experiments and analysis clarification to make the paper easier to follow and to clearly show our main achievements.*

Comment 1- In comparison with the previous reported porous carbon, what's the striking advance in fundamental understanding your carbon structures and what are the merits of the strategy in preparing the materials? In order to prove the advancement of this study, I recommend you to refer more papers published in recent years.

Response: *Thanks for the reviewer's recognition on our work and these appreciated questions. First, we have cited more recent published papers related to porous nano-carbon structures (ref. 1, 3, 6, 15, 16, 17, 18, 49, 50), and compared the merits of our methods with those reported ones, as indicated in the revised manuscript (lines 55-60; lines 72-76). We added more discussions in the revised manuscript to clarify the novelty of our research (lines 359-374). Combined with the sincere comments from the reviewer #3, we would like to clarify the novelty from the following aspect.*

The fabrication of conductive porous carbon nanofibers is a hot research area in materials chemistry. However, designing continuous CNFs with both high pore volume and large conductivity is quite difficult and is regarded as an open problem. The biggest difference between our PCNFs with the previously reported ones lies in that we increased the porosity and conductivity of PCNFs simultaneously while maintaining their structural integrity. Another novelty is that the solvent we used for dissolving the precursor materials was deionic water, which was very valuable for fabricating CNFs. While the traditional methods for fabricating PCNFs needed to use

large amounts of toxic solvents like DMF.

In this contribution, we solved these problems with a facile method in an environmental friendly manner by using water-soluble PVA as carbonized polymer and PTFE as pore induce polymer. The covalently functional groups of C-B, C-N and C-F increased the contents of SP² structures, rendering the CNFs with a high conductivity. While the evaporation of PTFE and boric acid created lots of continuous macro pores and meso-micro pores. The CNFs with different conductivities were prepared by varying the contents of B, F and N and optimizing pyrolysis temperatures (Figure S22). In addition to the advantages of controlling the conductivity and porosity, the versatility of the developed method realized the creation of interconnected macro-pores on nanoscale carbon fibers, and solved the defects of non-expansion or high cost in the conventional preparation of PCNF films.

Comment 2-The second point is about the flexibility of the porous carbon nanofibers. The authors stated that the pre-oxidation process could enhance the stability of the primary fibers, but the explanation of the phenomenon is not clear, you need to add more discussions. In addition to demonstrate the flexibility of the film by using some digital figures, you also need to test the flexible properties like stress-strain. Last, you should compare the flexibility of your materials with the reported ones, since it's amazing to maintain well of the porous structures.

Response: *We thank the reviewer for these appreciated questions. We have added more explanations on how the pre-oxidation process enhancing the stability of the primary fibers (Lines 119-124). Without using PTFE, the as-spun PVA nanofibers could not transform into CNFs after pyrolysis, since PVA started to melt at ~150 °C and flowed into a film at ~200 °C (Figure S2). Actually, once the content of PVA was larger than 80 wt.% in the PVA-PTFE system, the final products were not fibers, but like flat films (Figure S10 and S21). PTFE could prevent the diffusion of PVA at a higher temperature, ex. 280 °C. At this temperature, the chains of the PVA changed into stable conjugated C=C bonds, enhancing the stability of the final PCNFs.*

For the second question, we have added the stress-strain curve in the revised manuscript (Figure 3a and 3b), and the corresponding illustrations were added (lines 222-229). The PCNF films delivered a high Young's modulus of 429 MPa with a mechanical strength of 2.23 MPa and an extended strain of 0.52% before failure. In addition, the PCNFs exhibited a bending rigidity of only 13.1 mN, which was much smaller than the value of paper towel. Such mechanical performance made it possible

to apply these PCNFs into real applications for polluted gas adsorption, sewage disposal, liquid storage, energy storage and so on.

Comment 3-The third important point is about the conductivity of the porous carbon nanofibers. Is the conductivity from a single nanofiber or from the whole film? The resistance of such flexible (or freestanding) film will increase once its area is increased. You should test at least two different samples with different areas. You also should pay more attention to the root cause of the high conductivity.

Response: *Thanks for these appreciated questions. The demonstrated conductivity in this paper were from the whole film. The samples that were used for the conductivity testing had areas of $2 \times 2 \text{ cm}^2$ and $4 \times 4 \text{ cm}^2$. For both situation, we found the conductivity of the PCNF films were almost the same.*

*Combined with the reviewer #1, we would like to illustrate the high conductivity from the following three aspects. **First**, the high conductivity of 980 S/cm was achieved from the PCNFs that produced at 1200 °C. Such a high temperature rendered the PCNFs with a mixture of graphitic, turbostratic and amorphous structures. **Second**, from the cross-sectional SEM (**Figure S14**) and TEM (**Figure 2c**) images, our PCNFs had thorough porous structures. That means, each single CNF was composed by numerous thinner carbon sticks with only several nanometers. In our previously report (Nat. Commun. 2019, 10, 1458), we verified that such kind of CNF (with a diameter smaller than 30 nm) membranes had superior electrical conductivity of $\sim 750 \text{ S/cm}$. Of course, the deeply analysis for the conductivity enhancement need to be clarified in the future, and we are investigating the conductivity mechanism of such kind of PCNFs. **Thirdly**, unlike the conventional synthesis that rely on blending pitch or PAN with the doping elemental materials, we adopted BA, PVA and PTFE to fabricate PCNFs. Here, there were already a lot of C-F bonds in PTFE, while the crosslinking reactions between PVA and PTFE with BA as the crosslinking agent led to stable C-B bonds. We think there maybe some new mechanisms for forming C-N bonds in the PVA-PTFE based PCNFs under N_2 atmosphere. In the revised manuscript, we compared the conductivities of PCNFs that were prepared by varying the gas atmosphere (**Figure S22** and **S23**). Although the N_2 atmosphere could produce higher conductivity, the conductivity of the PCNFs that produced with Ar atmosphere were still much larger than the traditional CNFs that produced without using PTFE (eg. iScience 2019, 16, 122-132), indicating the C-F and C-B had great effects for improving the conductivity. the covalently functional*

groups of C-B, C-N and C-F increased the contents of SP² structures, rendering the CNFs with a high conductivity. We have added more discussions in the revised manuscript (lines 392-403).

Comment 4 The fourth important point is the wide applications of the porous carbon nanofibers. Firstly, the porous carbon membranes exhibited both hydrophobicity and rapid electrolyte wettability, can the authors explain the mechanisms of these two different phenomena? The insert figures in Figure 3d does not show the phenomenon you described and I recommend to re-edit this image. Secondly, it is interesting that the porous carbon materials had such a wide range of applications. Which of these applications do you think is more likely to be commercialized? I recommend to highlight its advantages and give more descriptions.

Response: *Thanks for these appreciated questions. For the first question, combined with the reviewer #1, we would like to illustrate these two different mechanisms as following. **First**, water droplets are objects with clearly defined surface and surface tension. When they contact the surfaces of nano-size structure, the air will absorb the voids of the nano-grooves and forms "air cushion" between the droplets and the surface, which can effectively prevents the water droplets from infiltrating. Superior hydrophobic surfaces can be prepared by two ways: one is to change the surface roughness and morphology of the materials; the other one is to modify low surface energy materials on the surface of the materials with a certain roughness. For one thing, the porous structure of the synthesized PCNFs increases the roughness of the surface, thus increasing the hydrophobic properties. For another, the F doping decreased the critical surface tension of the PCNF membrane. Specifically, the C-F group had a very low surface energy since the C-F bond had a high energy but a low polarization. F was regarded as an hydrophobic agent and the F-containing polymers were always applied for fabricating waterproof textiles. Therefore, the super hydrophobic property could be achieved by combining the F-doping with the surface roughening. **Second**, the structures of carbon materials would change when they were doped by heteroatoms, and the carbon surface would take polar groups. There were many oxygen atoms and other atoms including B, N and F in the PVA-based PCNFs, these polar groups had strong interactions, such as Van der Waals force, with the organic solvents of DME, EG and DOL. On the other hand, PCNFs are inorganic materials with large surface tension and high surface energy, which enhanced the wicking ability by organic liquids. We have added some more discussions in the*

revised manuscript (**Lines 257-263; lines 282-285**).

*For the second question, we have re-edited the previous Figure 3d as Figure 3e and revised the corresponding descriptions in the revised manuscript (**line 267**). For the third question, we think the applications of the PCNFs in energy storage are more likely to be commercialized since for energy storage devices, both the porosity and conductivity are essential. We have added more discussions in the revised manuscript (**lines 220-229; Lines 413-417**).*

Comment 5-Overall, this paper needs some refinements and more focus on the details about the mechanisms of porosity and high conductivity. It is also important to state over whether these structures can be reproducibly created and what's the production scale?

Response: *We thank the reviewer for these appreciated comments. We have carefully revised the whole manuscript and discussed more about the mechanisms of controlling the porosity and conductivity of the PCNFs (**lines 108-111; lines 119-124; lines 128-129; lines 131-144; lines 146-151; lines 154-156; lines 383-385; lines 392-403**). With respect to the reproducibility and large-scale production, two important issues should be carefully considered: solvents and fabrication process. Here, the solvent was deionized water. However, to make the PAN-based CNFs, the solvent was always toxic and expensive DMF. In addition, because of the simplicity of the self-assembly process of our methodology and the facile availability of the designed precursor solutions, we think scaling up the synthesis of PCNFs was possible. Actually, in our previous study, using our lab equipment (DXES-V spinning machine, SOF Nanotechnology Co., Ltd., China), we could easily obtain a uniform membrane with a large width of 120 cm. Therefore, we have confidence in the fabrication of such PCNF membranes with larger areas using more syringes and larger collectors. We have added these discussions in the revised manuscript (**lines 413-415**).*

Reviewer #3:

Overall comment:

The manuscript "Multifunctional Flexible Membranes from Sponge-Like Porous Carbon Nanofibers with Metallic Conductivity" by Dong et al. reports on a novel approach to fibrous carbon sponges with electric conductivity by electrospinning, followed by pyrolysis. The manuscript is surely of interest for the readers of Nature

Communication but requires major revision prior to publication due to some flaws and unclear information. It also appears that some more in depth analysis but somewhat less on application would have been better.

Response: *We thank the reviewer's recognition on our work; the comments are very helpful in improving our manuscript. On the basis of the sincere suggestions, we have appropriately revised the manuscript and added additional experiments and analysis clarification to make the paper easier to follow and to clearly show our main achievements. We hope that the reviewer finds our responses satisfactory and convincing. The responses to comments are listed below:*

Comment 1-A general remark: page numbers and line numbers would make the review and revision process easier.

Response: *Thanks for this sincerely suggestion. We have added page numbers in the revised manuscript.*

Comment 2-The reported electrical conductivity is by two orders of magnitude below the conductivity of metals. Therefore, the term “metal-like” conductivity cannot be applied.

Response: *Thanks for this respect suggestion. We have changed the “metal-like” conductivity as “high” conductivity in the revised manuscript.*

Comment 3-Refer to the sentence: “The concept of freestanding electrode has been reported in a number of papers.” Papers should be cited here. A simple Google search on sponges or electrospinning and electric conductivity brings many papers for porous scaffolds including carbons with high conductivity and even higher, truly metal-like conductivity. The authors should cite these papers and discuss the novelty of their work in the light of relevant literature.

Response: *Thanks for these appreciated questions. We have cited more related papers in the revised manuscript according to the suggestion (ref. 15-18). For the second question, we have added these discussions in the revised manuscript (lines 55-60; lines 359-374) to clarify the novelty. There were indeed some reports on electrospun CNFs with high conductivity. However, designing continuous CNFs with both high pore volume and large conductivity is quite difficult and is regarded as an open problem. The biggest difference between our PCNFs with the previously reported ones lies in that we increased the porosity and conductivity of PCNFs*

simultaneously while maintaining their structural integrity. Another novelty is that the solvent we used for dissolving the precursor materials was deionized water, which was very valuable for fabricating CNFs. While the traditional methods for fabricating PCNFs needed to use large amounts of toxic solvents like DMF.

*The conductivity of PCNFs is generally inversely proportional to the pore volumes. For example, the sp^2 -hybridized carbon nanomaterials generally had high electrical conductivity but limited pore volumes, while porous carbon nanomaterials could afford abundant pore structures, but the amorphous or sp^3 -hybridized carbon structures in these nanostructured carbon caused poor electrical conductance. In this contribution, we solved these problems with a facile method in an environmental friendly manner by using water-soluble PVA as carbonized polymer and PTFE as pore induce polymer. The covalently functional groups of C-B, C-N and C-F increased the contents of SP^2 structures, rendering the CNFs with a high conductivity. While the evaporation of PTFE and boric acid created lots of continuous macro pores and meso-micro pores. The CNFs with different conductivities were prepared by varying the contents of B, F and N and optimizing pyrolysis temperatures (**Figure S22**). In addition to the advantages of controlling the conductivity and porosity, the versatility of the developed method realized the creation of interconnected macro-pores on nanoscale carbon fibers, and solved the defects of non-expansion or high cost in the conventional preparation of PCNF films.*

Comment 4-Refer to the sentence: “At this temperature, the semi-fluid PVA had a...” Semi-fluid is too qualitative for any reproduction. Please give quantitative data for the viscosity of the formulation.

Response: *Thanks for this appreciated question. We have added more discussions and experiments to verify the fluidity of the PVA particles at high temperatures of 180 °C and 210 °C in the revised manuscript (**Figure S2, lines 120-124**). At 180 °C, the PVA had a higher viscosity of 8989 cp, while at 210 °C, it exhibited a low viscosity of 2794 cp. Of note, once turning off the heater, the PVA could quickly form a film. The color of the primary PVA particles was yellow, when being heated in air, it became red.*

Comment 5-Refer to the sentence: “The PCNFs were doped with B, F and N (Fig. 1f). “Here, sound elemental analysis is missing, e. g. by EDX. Otherwise this remains mere speculation.

Response: *We thank the reviewer for this good question. We fully understand the reviewer’s concerns about the B-F-N triply doping. In the manuscript, the EDS (Figure 1f) was tested with a very low speed along with the TEM figures, in which the distribution of N coincided well with the porous structures of the single PCNF. We have tried to test EDX, but the results were not good due to the low sensitivity of EDX. To further confirm this, we tested XPS in the revised manuscript, as shown in Figure S8. There were C, O, N, B and F in the PCNFs. The atomic percents of C, O, N, B and F were 94.33%, 3.15%, 1.25%, 0.93% and 0.34%, respectively.*

Comment 6-Fig. 1. The image of the fibrous sponge-like fiber is too speculative. Clearly, a cross-sectional SEM image is missing in order to verify that pores are really continuously.

Response: *Many thanks for these appreciated comments. We have taken some new SEM images and changed Figure 2b in the revised manuscript. In addition, we have added four additional cross-sectional SEM images in the revised supplementary information (Figure S14).*

Comment 7-Fig. 2e. Nice demonstration of robustness but here quantitative mechanical data are absolutely essential.

Response: *We thank the reviewer for this appreciated comment. We have added these data in the revised manuscript (Figure 3a and 3b), and the corresponding illustrations were added (lines 222-229). The PCNF films delivered a high Young's modulus of 429 MPa with a mechanical strength of 2.23 MPa and an extended strain of 0.52% before failure. In addition, the PCNFs exhibited a bending rigidity of only 13.1 mN, which was much smaller than the value of paper towel. Such mechanical performance made it possible to apply these PCNFs into real applications for polluted gas adsorption, sewage disposal, liquid storage, energy storage and so on.*

Comment 8-Fig. 3a. at presence too speculative – see above

Response: *Thanks for this sincere comment. We agree with the reviewer and have removed this figure in the revised manuscript.*

Reviewers' comments:

Reviewer #1 (Remarks to the Author):

1. It is claimed "Figure S15. TEM characterizations of the graphitic carbon structures in PCNFs. There were obviously layered carbon structures with an average inter-planar distance of $\sim 0.41\text{nm}$ ". The inter-planar distance of $\sim 0.41\text{nm}$ for PCNFs is far from 0.334 nm of graphite, indicating the crystalline of PCNFs is amorphous, not graphitic. Also, "Figure S7. XRD of the PCNF-1200 oC, which indicated a high graphitization degree of carbon in the PCNFs" is not correct. The wide 002 peak also characterize amorphous carbon.
2. The ultrahigh conductivity was attributed to the elemental doping of N, B and F. However, XPS indicates the element content of N, B and F are only 1.25%, 0.93% and 0.34%, respectively. How this low content element can endow the PCNFs so large conductivity (980 mS/cm)?
3. Similar to above question, the ultrahigh adsorption ability of PCNFs to ethanol DME, DOL, is also explained by the heteroatom doping, but the element content of N, B and F is very low.
4. The authors explained the pore creation mechanism as "The micro and macro phase separations brought a homogeneous pore distribution, in which small molecule (BA and PVA) pyrolysis created meso and micro pores, while the decomposition of large PTFE NPs left continuous macro pores." However, PVA, BA and PTFE were formed homogeneous spinning sol of PVA-BA-PTFE before electrospinning process. So no large PTFE NPs (120 nm) are existed, how did it form macropores?
5. Being used in supercapacitor, the energy density is calculated to be 42.77 Wh Kg^{-1} from charge-discharge curves. The charge-discharge curves and the capacitance at 1 A/g should be given. The micro-meso-macroporous structure make the PCNF should have good rate performance, but the Impedance spectra, charge-discharge and CV can not support this. 50 mV s^{-1} is not a high scan rate and the capacitance at enhanced current density are suggested to be added.
6. The carbon yield was only 4.75%, which is a very low value.
7. In Figure S23, is "Pyrolysis in air" correct? Heating at $800\text{-}1000\text{ }^{\circ}\text{C}$, the PCNF will burned off.

Reviewer #2 (Remarks to the Author):

The authors's answer to the comments was satisfied.

Reviewer #3 (Remarks to the Author):

The authors have answered all my concerns in a satisfactory way. The manuscript is cleared according to my concerns.

Just a very minor issue: Check spelling of nanofibers and iSciece Ref. 17

Point-by-point responses to Reviewers' critiques

Manuscript ID: NCOMMS-19-19429A

Note: All critiques are highly appreciated. The changes are **highlighted in blue** in the **revised Manuscript** and **revised Supplementary Information** for review as suggested.

Comments from reviewers:

Reviewer #1:

Point-by-point response

Comment 1- It is claimed “Figure S15. TEM characterizations of the graphitic carbon structures in PCNFs. There were obviously layered carbon structures with an average inter-planar distance of ~0.41nm”. The inter-planar distance of ~0.41nm for PCNFs is far from 0.334 nm of graphite, indicating the crystalline of PCNFs is amorphous, not graphitic. Also, “Figure S7. XRD of the PCNF-1200 °C, which indicated a high graphitization degree of carbon in the PCNFs” is not correct. The wide 002 peak also characterize amorphous carbon.

Response: *Many thanks for this valuable comment, and we truly appreciate this critical question. **First**, we are sorry for making such confusion in the last edition, we have removed the word of “graphitic” in Figure S15. **Secondly**, we double-checked TEM for the PCNFs, and the new result is shown in Figure S15. **Thirdly**, we retested XRD for the PCNFs by grinding the PCNF film into powders, and the new result is shown in Figure S7.*

In this manuscript, we claimed the carbon structure as “a mixture graphitic, turbostratic and amorphous”. As shown in the TEM figures, graphitic structures are observed at the edge of the fibers and near the pores, while the other parts are amorphous structures. For the graphitic structure, the inter-planar distance is around 0.37 nm. Here, we want to claim that the highly graphitized carbon lattice of 0.334 nm requires very high carbonization temperature. Our PCNFs had a mixture of graphitic, turbostratic and amorphous carbon domains with different ratios depending on the degree of graphitization. There are lots of literature stated that the d-spacing of the graphitic carbon could be between 0.334-0.43 nm. For example, in

this paper (*Energy and Environment Sciences*, 2014, 7, 2689-2696), the authors' results indicated that "The d-spacing of the graphene sheets in the nanofibers, corresponding to the (002) plane of the carbon, is in the range of 0.34–0.43 nm". We have revised the manuscript in the new version (**Lines 171-174**)

For the XRD spectra, the peaks of (002) and (100) indicate that there are graphitic carbon structures in the PCNFs. Of course, the broadening of (002) peak characterizes the existence of amorphous carbon within PCNFs. We hope the reviewer find our analysis are satisfactory and convincing.

Figure S15. HRTEM characterization of the PCNF-1200 °C.

Figure S7. XRD spectra of the PCNF-1200 °C.

Comment 2- The ultrahigh conductivity was attributed to the elemental doping of N, B and F. However, XPS indicates the element content of N, B and F are only 1.25%, 0.93% and 0.34%, respectively. How this low content element can endow the PCNFs so large conductivity (980 mS/cm)?

Response: *Thanks for this appreciated question. We fully understand the reviewer's concerns about the conductivity. After checking some papers related to element doped CNFs, we found that a low element doping content (< 4%) could also enhance the electrical conductivity greatly (Progress in Materials Science, 2013, 58, 565; Solid state Sciences, 2011, 13, 1459; Carbon Letters, 2018, 27, 1; et al.). To find the reason, we did a lot more experiments about the conductivity, as shown in the following Figure 1 and Figure 2. We would like to illustrate the conductivity from two aspects.*

First, from Figure 2, even only with F-doping, the PCNFs demonstrated high conductivity. While for the conventional CNFs synthesized from PAN, the conductivity was generally smaller than 20 S/cm. We think the method of using PVA as the carbon precursor and PTFE as the pore-inducer may have great influence on the conductivity. As claimed in the manuscript, at 280 °C, PVA flows onto the surface of PTFE particle, which has a high decomposition temperature of ~600 °C. That means, during the carbonization process, PVA and PTFE have sufficient contact, which facilitates the formation of conjugated C=C and C-F bonds. However, we cannot verify the effects of PTFE doping on the conductivity, since without using PTFE, we could not obtain

CNFs due to the low thermal stability of PVA.

Second, the PCNFs had hollow porous structures. Such highly porous structures divided the CNFs into many thin carbon layers, like carbon nanotubes. We are not sure, but at least from the experiments, such thin carbon layers enhanced the conductivity, since the different micro-size carbon fibers that fabricated with similar methods (check it from comment 6) had a low conductivity of ~ 112 S/cm. Actually, some reports show that CNFs had very high conductivity of larger than 10000 S/cm. For example, in this paper (Carbon 2016, 99, 407-415) the microscale carbon fibers had a conductivity over 12000 S/cm. It is worth to note that the high conductivity of 980 S/cm was achieved from the PCNFs that produced at 1200 °C. However, with a low calcination temperature, the conductivity quickly decreased. We hope the reviewer find our analysis are satisfactory and convincing.

Figure 1, SEM characterization of PCNFs that synthesized under different conditions. We call (A-C) as F- and B- doped PCNFs, (D-F) as F- doped PCNFs, (G-I) as F- and N- doped CNFs, and (J-L) as B-, F- and N- doped PCNFs.

Figure 2, conductivity of the synthesized PCNFs.

Comment 3- Similar to above question, the ultrahigh adsorption ability of PCNFs to ethanol DME, DOL, is also explained by the heteroatom doping, but the element content of N, B and F is very low.

Response: We thank the reviewer for this valuable comment, we would like to answer this question from the following two aspects.

First, most reported CNFs, especially that with rich surface functional groups (ex. -O group), have good organic liquid wettability, since CNFs are typical and famous electrode materials or conductive agents in electrodes, which have good absorption capacity for organic electrolytes.

*Second, for our PCNFs, the heteroatom doping is an addition of surface functional groups on the CNFs, which can enhance the organic solvent wicking speed. But we believe that it is the pore volumes that determine the ultrahigh adsorption ability. We have revised the manuscript and make it more clear (**Lines 282-284**). We hope the reviewer find our analysis are satisfactory and convincing.*

Comment 4- The authors explained the pore creation mechanism as “The micro and macro phase separations brought a homogeneous pore distribution, in which small molecule (BA and PVA) pyrolysis created meso and micro pores, while the

decomposition of large PTFE NPs left continuous macro pores.” However, PVA, BA and PTFE were formed homogeneous spinning sol of PVA-BA-PTFE before electrospinning process. So no large PTFE NPs (120 nm) are existed, how did it form macropores?

Response: *We thank the reviewer for this valuable comment. The PTFE we used in this experiment was an water emulsion (Line 439), in which PTFE nanoparticles suspended in water. PTFE is very difficult to dissolve in the conventional solvents. According to our knowledge, only perfluoroeicosane can dissolve PTFE.*

Comment 5- Being used in supercapacitor, the energy density is calculated to be 42.77 Wh Kg⁻¹ from charge-discharge curves. The charge-discharge curves and the capacitance at 1 A/g should be given. The micro-meso-macroporous structure make the PNCf should have good rate performance, but the Impedance spectra, charge-discharge and CV can not support this. 50 mV s⁻¹ is not a high scan rate and the capacitance at enhanced current density are suggested to be added.

Response: *Many thanks for these valuable questions. First, the charge-discharge curves at 1 A/g were in the last version of manuscript. The capacitance at 1 A/g is 25.2 F/g (Lines 321-323). For the second comment, we have added more scan rates from 100 to 500 mV/s in Figure 4C and the related discussions have been added in the revised manuscript (Lines 312-318).*

Comment 6- The carbon yield was only 4.75%, which is a very low value.

Response: *We thank the reviewer for this insightful comment. Indeed, 4.75% is not a high value in comparison with the traditional polymer precursors of carbon fibers, for example, the carbon yield of PAN is generally 25%-50%, and the carbon yield of PVP (polyvinyl pyrrolidone) is generally 5%-15%.*

In this manuscript, we focus on the novel method for creating PCNFs with hollow porous structures. And this method can be applied into the PAN-based CNFs. Very recently, we have successfully developed highly porous micro-carbon fibers with stronger strength and larger specific surface areas, as shown in the following figure,. These results indicate that the method is scalable. We hope the reviewer find our analysis are satisfactory and convincing.

Comment 7- In Figure S23, is “Pyrolysis in air” correct? Heating at 800-1000 °C, the PCNF will be burned off.

Response: *Thank the reviewer for this comment. We did not write “Pyrolysis in air”, we wrote ‘Pyrolysis in Ar (argon)’, as you can double check in Figure S23. Here, we used the experiments to verify the effect of N-doping on the conductivity of PCNFs.*

Reviewer #2:

Comment: The authors' answer to the comments was satisfactory.

Response: *We thank the reviewer for the recognition on our work and thanks for the time reviewing this manuscript.*

Reviewer #3:

Comment: The authors have answered all my concerns in a satisfactory way. The manuscript is cleared according to my concerns.

Just a very minor issue: Check spelling of nanofibers and iScience Ref. 17

Response: *We thank the reviewer for the recognition on our work and thanks for the time reviewing this manuscript. We have contacted a professional language*

editing service to correct spelling, tense, punctuation, and capitalization errors. Awkward sentences and phrases have been removed to improve the clarity of the paper. Some sentences were rearranged or rewritten to improve reader understanding.

REVIEWERS' COMMENTS:

Reviewer #1 (Remarks to the Author):

The manuscript can be accepted now.